# Single-cell analysis highlights differences in druggable pathways underlying adaptive or fibrotic kidney regeneration

Michael S. Balzer [1,2], Tomohito Doke [1,2], Ya-Wen Yang[1,2], Daniel L. Aldridge [3], Hailong Hu [1,2], Hung Mai[1,2], Dhanunjay Mukhi[1,2], Ziyuan Ma [1,2], Rojesh Shrestha[1,2], Matthew B. Palmer[4], Christopher A. Hunter [3] & Katalin Susztak [1,2,5✉]

The kidney has tremendous capacity to repair after acute injury, however, pathways guiding adaptive and fibrotic repair are poorly understood. We developed a model of adaptive and fibrotic kidney regeneration by titrating ischemic injury dose. We performed detailed biochemical and histological analysis and profiled transcriptomic changes at bulk and single-cell level (> 110,000 cells) over time. Our analysis highlights kidney proximal tubule cells as key susceptible cells to injury. Adaptive proximal tubule repair correlated with fatty acid oxidation and oxidative phosphorylation. We identify a specific maladaptive/profibrotic proximal tubule cluster after long ischemia, which expresses proinflammatory and profibrotic cytokines and myeloid cell chemotactic factors. Druggability analysis highlights pyroptosis/ferroptosis as vulnerable pathways in these profibrotic cells. Pharmacological targeting of pyroptosis/ferroptosis in vivo pushed cells towards adaptive repair and ameliorates fibrosis. In summary, our single-cell analysis defines key differences in adaptive and fibrotic repair and identifies druggable pathways for pharmacological intervention to prevent kidney fibrosis.

---

[1] Renal, Electrolyte, and Hypertension Division, Department of Medicine, Perelman School of Medicine, University of Pennsylvania, Philadelphia, PA 19104, USA. [2] Institute for Diabetes, Obesity and Metabolism, Perelman School of Medicine, University of Pennsylvania, Philadelphia, PA 19104, USA. [3] Department of Pathobiology, School of Veterinary Medicine, University of Pennsylvania, Philadelphia, PA 19104, USA. [4] Department of Pathology and Laboratory Medicine, Perelman School of Medicine, University of Pennsylvania, Philadelphia, PA 19104, USA. [5] Department of Genetics, Perelman School of Medicine, University of Pennsylvania, Philadelphia, PA 19104, USA. ✉email: ksusztak@pennmedicine.upenn.edu

Every year, more than 13 million people suffer from acute kidney injury (AKI). The kidney tubules have tremendous capacity to regenerate following AKI and most cases resolve via adaptive regeneration, however, in some patients AKI leads to long-term fibrosis and chronic kidney disease (CKD). CKD is the fourth fastest growing cause of death, affecting more than 850 million people worldwide[1]. Although hard to estimate, AKI is one of the top contributors to CKD[2]. AKI increases the risk for progression to CKD (HR 2.67, 95% CI 1.99–3.58), end stage renal disease (HR 4.81, 95% CI 3.04–7.62), and death (HR 1.80, 95% CI 1.61–2.02)[3]. Understanding the pathomechanism of adaptive and fibrotic repair is therefore critical.

During the last decade, we made important progress in our understanding of adaptive kidney repair and regeneration. Multiple studies have focused on identifying the source of progenitor or stem cells in the kidney following AKI. While early work indicated a potential role for mesenchymal stem cells, rigorous linage tracing experiments demonstrated a key role of epithelial cells in regeneration[4,5]. Several reports indicate that Sox9-positive cells expand, proliferate, and differentiate following injury[6–8]. Other studies have identified Lgr4 and Lgr5-positive precursor cells[6,9,10]. Activation of classic developmental pathways such as Wnt and Notch plays an important role in lineage decision and transit amplification promoting regeneration[11], however, sustained activation of these pathways inhibits full differentiation of epithelial cells. Spatial and temporal organization of developmental signals are critical for proper regeneration.

Several events have been identified in the AKI-to-CKD transition. Compromised microvascular integrity, tubular epithelial cell damage[12], interstitial inflammation[13,14] and myofibroblast recruitment[15], and are key factors of maladaptive repair[16–18]. However, the exact mechanisms have not been fully evaluated. One of the most important changes observed in CKD is the influx and expansion of immune cells. Immune cells likely secrete cytokines to heal the epithelium but could also play a critical profibrotic role[19,20]. Immune cell subtypes and their activation are poorly characterized in the context of AKI-to-CKD transition.

A key bottleneck in the understanding of adaptive and maladaptive regeneration has been the limited insight into temporal and cell-specific genome-wide gene expression changes to define signals that initiate and drive cell differentiation and cell–cell interactions. Single-cell RNA sequencing (scRNA-seq) has fundamentally improved our understanding of kidney disease development in mice and humans[21–33]. Several studies have analyzed maladaptive regeneration following AKI and identified a variety of pathways. However, these studies failed to compare adaptive and maladaptive regeneration to comprehensively differentiate pathways attributable to successful repair and fibrosis[34]. In addition, only one study performed follow-up validation experiments confirming the role of specific pathways[35].

Here we develop a model of adaptive and maladaptive regeneration and directly compare single-cell transcriptomic profiling in these conditions by defining cell-autonomous and cell–cell interaction changes, identifying and validating factors that drive maladaptive regeneration. We show that maladaptive (fibrotic) repair is associated with activation of inflammatory cell death pathways (pyroptosis and ferroptosis) inducing the influx and activation of immune cells.

## Results

### Single-cell transcriptome dynamics of adaptive and maladaptive regeneration following bilateral renal ischemia. To understand differences between renal regeneration and fibrosis we established a model of adaptive and maladaptive repair by carefully titrating renal ischemia time in mice. We analyzed

changes following short (23 min) or long (30 min) bilateral ischemia and collected samples 1, 3, and 14 d post-ischemia (Fig. 1a, Supplementary Dataset 1, "Methods"). Kidney function measured by iSTAT serum creatinine and blood urea nitrogen (BUN) showed AKI peaking on day 1 (BUN 149 and 83 mg/dL, creatinine 1.2 and 0.5 mg/dL after long and short ischemia reperfusion injury (IRI), respectively). These values were similar to functional impairments observed in recently published studies[34]. By day 14, creatinine levels returned to baseline in both groups, while BUN levels remained elevated in the long ischemia group (Figs. 1b, S1a). Blinded histopathological scoring confirmed similar early structural changes on day 1 and 3 post-IRI in both groups and sustained acute tubular injury 14d post-IRI in mouse kidneys. Mice with prolonged kidney ischemia developed fibrosis by day 14 following ischemia (Fig. 1b–e).

We performed bulk and scRNA-seq from the same kidneys on 2 mouse samples at each time point, yielding a total of 18 samples (n = 6 long IRI, n = 6 short IRI, and n = 6 controls). Marker genes of fibrosis such as Fn1, Tgfb1, Col1a1, and Col3a1 were higher in long ischemia samples in bulk RNA-seq data (Fig. 1f, Supplementary Dataset 2). We also performed scRNA-seq, aggregated transcriptomes of high-quality single cells into a single dataset following batch integration with LIGER[36], and retained 113,579 high-quality cells after rigorous filtering based on UMI counts, mitochondrial percentage, doublet removal, and ambient RNA and batch effect correction (Figs. S1b–d, S2a, "Methods"). Our analysis indicated 18 clusters. We next performed analysis of differentially expressed genes (DEGs) for each cluster and identified all major kidney cell types, including endothelial cells, podocytes, tubule cells, and a variety of immune cells (Fig. 1g, h and Supplementary Dataset 3).

### Cell fraction changes during repair and fibrosis. To gauge the effect of the degree of injury on cell fraction changes, we analyzed changes in cell proportion in single-cell and bulk RNA-seq samples. We observed considerable differences in cell fractions that correlated both with the degree of ischemia and the timing following injury (Fig. S2b and Supplementary Datasets 4, 5). Immune cells constituted 10, 45, and 66% of analyzed cells in kidneys of control, short, and long IRI samples in the single-cell dataset; proximal tubule (PT) cell fraction decreased to 54, 12, 8% in control, short, and long IRI models (Fig. 1i, j). We also analyzed changes by cell density, which has an advantage of potentially avoiding artificially binning the cells into clusters. Tubule cell density was highest in short IRI and control samples (Fig. 1k). This confirmed the markedly higher immune cell density in the long IRI samples. Amongst the immune cells, we observed a greater increase in myeloid cells, the fraction of which correlated with ischemia dose (Figs. 1j, S2c).

As the single-cell analysis is sensitive to cell drop-out, we confirmed cell fraction changes by in silico deconvolution of bulk RNA-seq samples. We corroborated lower PT cell and higher myeloid cell fractions in IRI samples (Fig. S3a, b). Using a previous single cell dataset from our group that is known to be rich in immune cell subtypes as a reference for deconvolution[30], we also confirmed CD4 and CD8 T cell as well as macrophage, neutrophil, and basophil infiltration with increasing injury dose (Fig. 2a, b). Myeloid and PT cell changes were greater as time progressed after injury (Figs. 2b, S3b, S4a, b). Accordingly, myeloid and PT cell-specific genes were among the top-loading genes contributing most to principal component heterogeneity (Fig. S5a, b). Bulk RNA-seq confirmed that genes differentially expressed in repairing vs fibrotic samples could be deconvoluted to PT and myeloid cells, respectively (Fig. S5c–e), indicating that cell heterogeneity was a likely major contributor of bulk gene expression changes.

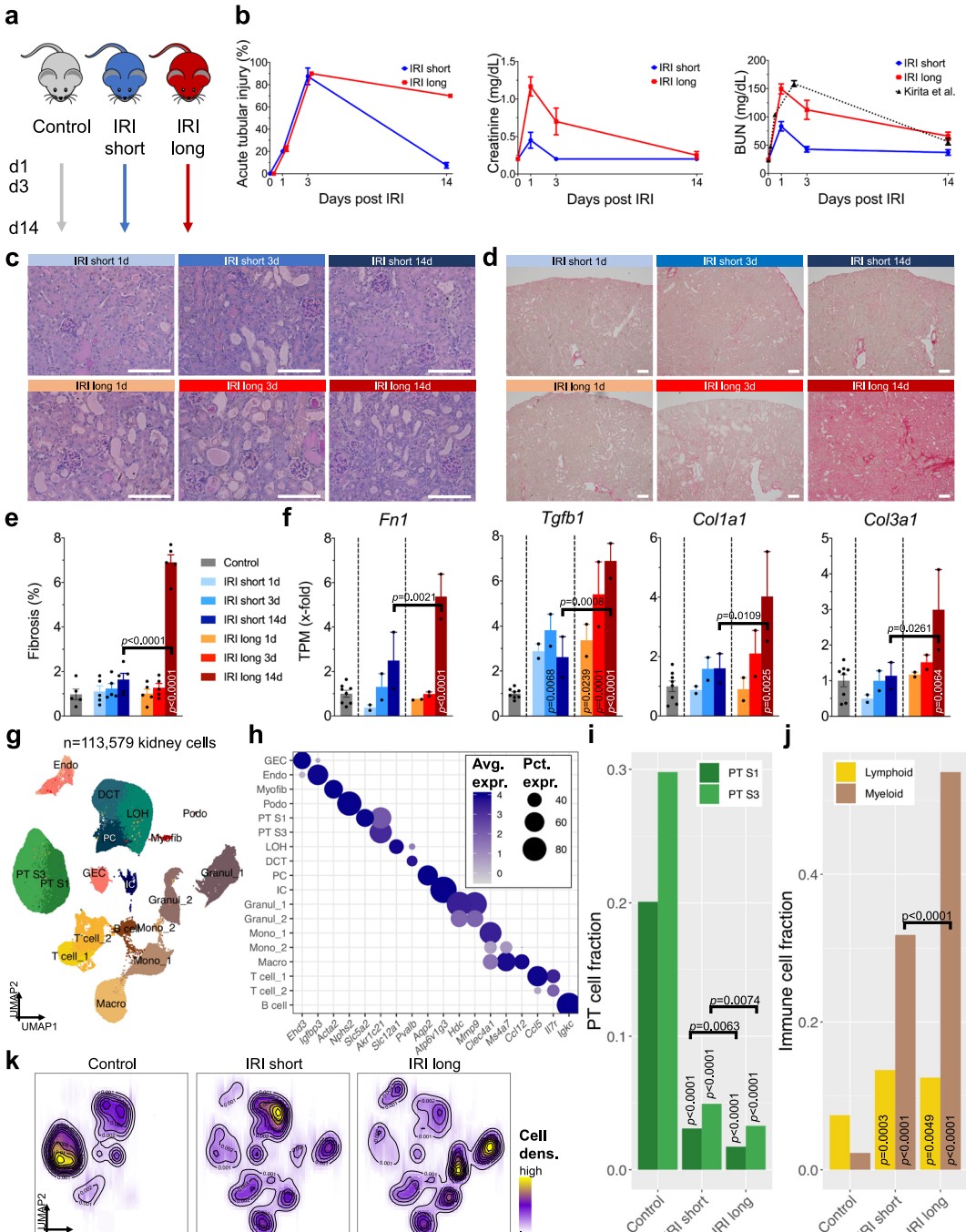

**Fig. 1 Single-cell transcriptome dynamics of adaptive and maladaptive regeneration following bilateral renal ischemia. a** Experimental setup. Male C57BL/6 mice were subjected to short and long ischemia and followed for 1, 3 and 14 d (*n* = 12). Bulk and scRNA-seq was performed in *n* = 2 mice per condition and time point and compared to *n* = 6 controls; artwork own production and from https://smart.servier.com, license https://creativecommons.org/licenses/by-sa/3.0/). **b** Histopathological evidence of acute tubular injury (left panel), blood creatinine (middle panel) and BUN changes (right panel) 1, 3, and 14 d post-IRI. BUN of long IRI was comparable to previously published data (Kirita et al.); means ± SEM. Representative light microscopy of Periodic acid-Schiff (**c**) and Sirius red (**d**)-stained kidneys; scale bars = 50 μm. **e** Fibrosis quantification from (**d**); means ± SEM; *p* values are given for one-way ANOVA comparisons between Control and IRI groups and between short and long IRI groups, respectively. **f** Bulk RNA-seq analysis of fibrosis markers *Fn1, Tgfb1, Col1a1*, and *Col3a1*; means ± SEM; *p* values are given for one-way ANOVA comparisons between Control and IRI groups and between short and long IRI groups, respectively. **g** UMAP projection of 113,579 cells (*n* = 6 controls, *n* = 6 short, *n* = 6 long IRI samples) passing rigid quality control filtering (nFeatures > 200 and < 3000, mt % < 50, doublet removal) and after dataset integration with LIGER, yielding 18 cell clusters: GEC glomerular endothelial cell; Endo endothelial cell; Myofib myofibroblast; Podo podocyte; PT S1 proximal convoluted tubule; PT S3, proximal straight tubule; LOH, loop of Henle; DCT, distal convoluted tubule; PC, principal cell; IC, intercalated cell; Granul, granulocyte; Mono, monocyte; Macro, macrophage; T cell; B cell. **h** Cell type-specific expression of marker genes for manually annotated clusters. Dot size denotes percentage of cells expressing the marker. Color scale represents average gene expression values. **i, j** Fractions of PT (**h**) and immune cell (**i**) clusters stratified by treatment group (Control; IRI short; IRI long). Statistical significance for comparisons was derived using differential proportion analysis, with a mean error of 0.1 over 100,000 iterations; *p* values are given for comparisons between Control and IRI groups and between short and long IRI groups, respectively. **k** Changes in cell type proportions between treatment groups (Control; IRI short; IRI long), visualized as cell density.

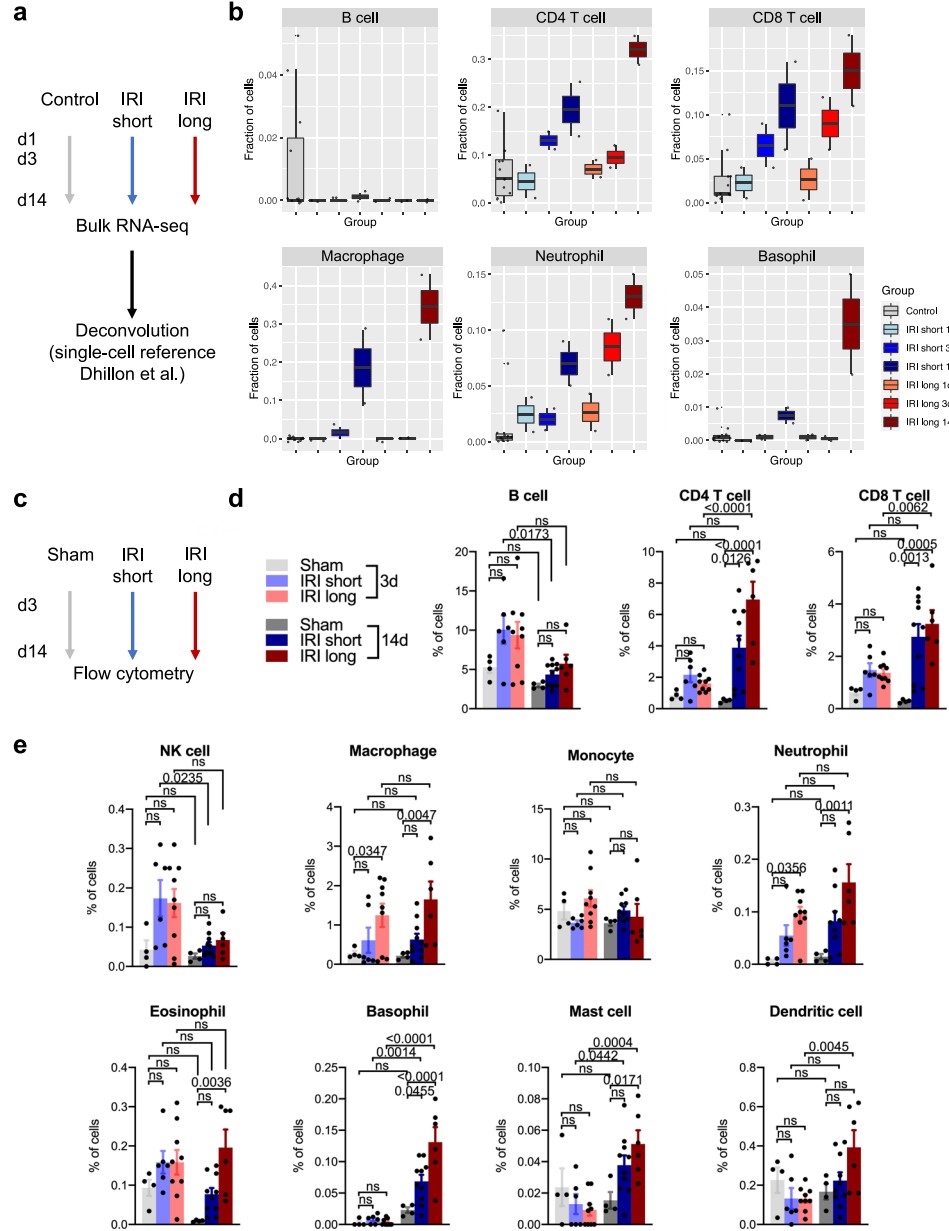

**Fig. 2 Bulk deconvolution and flow cytometry confirm lymphoid and myeloid cell infiltration in maladapted kidneys. a** Kidney bulk RNA-seq data were in silico deconvoluted using a previously published kidney single cell reference that is known to be rich in heterogeneous immune cell populations (Dhillon et al.). **b** Quantification of cell fractions after bulk RNA-seq deconvolution, displayed as Tukey box plots. **c** Flow cytometry was performed in an independent set of experiments; the gating strategy is shown in Fig. S6. **d, e** Flow cytometry quantification of lymphoid (**d**) and myeloid (**e**) cells representing $n = 4$ independent experiments. Increasing doses of ischemic injury resulted in significant increases in lymphoid and myeloid cell infiltration; means ± SEM; *p* values for one-way ANOVA (Holm–Sidak corrected); light coloring in bar graphs denotes 3 d time points, full coloring denotes 14 d time points; NK, natural killer.

To investigate whether dissociation bias due to single-cell preparation protocols might have skewed cell proportions we sought to validate immune cell fraction changes by an orthogonal method. As immune cells are defined by their cell surface markers, we chose flow cytometry as the gold standard of quantitation. We performed flow cytometry on mouse kidneys in an additional set of experiments to quantify immune cells such as lymphocytes (B, CD4, and CD8 T cells) and myeloid cells (macrophages, neutrophils, basophils, eosinophils, mast cells, and dendritic cells) infiltrating the kidney 3 and 14 d after sham, IRI short, and IRI long procedures, respectively (Figs. 2c, S6 and Supplementary Dataset 6). While early (3 d) after IRI, severe

injury significantly increased only macrophage and neutrophil influx, we observed ischemia dose-dependent increases of CD4 and CD8 T cells as well as pronounced increases of myeloid cell (macrophage, neutrophil, basophil, eosinophil, mast cell, dendritic cell) fractions 14 d after injury (Fig. 2d, e), thus independently validating cell fraction changes from our transcriptomic and deconvolution data. Again, flow cytometry highlighted the proinflammatory nature of the long IRI model with profound immune cell influx 14 d after injury.

In summary, renal IRI was characterized by marked changes in cell fractions including a dose-dependent, progressively lower PT cell and higher myeloid cell number.

**Profibrotic PT cells accumulate during maladaptive repair**. As PT cells showed some of the most dramatic changes, we focused our attention on these cells. We subclustered all 28,385 PT cells and identified 11 distinct subgroups (Figs. 3a, S7a). Three sub-clusters expressed classic markers of proximal convoluted tubule (S1) (distinguished by *Slc5a2, Slc5a12,* and *Gatm*). Two sub-groups expressed S3 proximal straight tubule (PST) markers (*Slc22a30, Atp11a, Inmt*). One subgroup expressed S2 PST marker *Slc22a6.* We identified 3 injured PT subclusters expressing either *Vcam1, Havcr1* and *Krt20,* or *Nupr1* (Figs. 3b, S7b–d and Supplementary Dataset 7). These injured PT subclusters were observed by previous studies[34,37,38]. In addition, we identified a proinflammatory subcluster expressing a variety of chemokines (*Il1b, Cxcl2, Ccl3, Tyrobp,* and *C3*).

We observed significant cell heterogeneity among PT cells (Fig. S7e, f). Control and IRI short samples showed the highest cellular densities in healthy PT clusters (Fig. 3c). Injured PT cells were higher on day 1 following injury; we found that the proinflammatory subcluster was present almost exclusively at later stages of the long IRI group (Figs. 3c, d, S7c, d), therefore we called this cluster maladaptive or profibrotic. This profibrotic cluster showed very strong IRI score enrichment (Fig. S7g–j), which we established using the top 100 IRI and top 100 Control DEGs (Fig. S7g, "Methods") and validated in published single-cell renal IRI data (Fig. S7k). To validate the maladaptive/profibrotic cluster, we analyzed our own (obtained from the same samples) and external bulk RNA-seq datasets and confirmed the presence and enrichment of this maladaptive cluster 14 d post long IRI (Figs. 3e, S8a, b)[37]. Moreover, maladaptive PT genes were enriched among PT cells of fibrotic kidneys in a unilateral ureteric obstruction (UUO) scRNA-seq dataset (Fig. 3f).

To further validate the presence of the maladaptive/profibrotic cluster, we analyzed previously published kidney IRI single-cell data[34]. Integration of all kidney cells from both datasets ($n = 240,043$) demonstrated highly conclusive overlap of cluster identities of all kidney cell types (Fig. 3g). We were intrigued to see considerable overlap of gene expression signatures of our injured and profibrotic/maladaptive cluster (Figs. 3h, i, S8c–e).

In summary, we identified heterogeneous PT subgroups, such as injured and maladaptive PT cells. We also validated their presence in a second kidney fibrosis model (UUO) and in previously published bulk and single-cell datasets.

**PT trajectories of adaptive and maladaptive repair**. To understand cell differentiation driving adaptive and maladaptive PT cell repair, we first defined PT cell lineage relationships using pseudotemporal cell ordering by Gaussian mixed modeling (GMM) (Fig. S9a, b, "Methods"). In a randomly down-sampled dataset ($n = 1650$ PT cells) we retrieved two distinct trajectories, both arising from cells injured early in the time course (Fig. 4a). Starting at 1d post-IRI, short IRI samples progressed along lineage 1 towards healthy (control) PT, indicating successful PT repair (Fig. 4b). Long IRI samples branched off along lineage 2 towards an endpoint formed by maladaptive profibrotic cells (Figs. 4b, S9c). RNA velocity analysis confirmed results obtained via GMM clustering (Fig. 4c) and IRI scoring showed the highest degree of injury at the endpoint of lineage 2 (Fig. 4d). We confirmed our trajectory results using Monocle2, Monocle3, and Slingshot trajectory analysis tools, indicating the robustness of transcriptomic relationships between adaptively repaired and profibrotic PT cells (Fig. S9e–j, "Methods"). Next, we examined DEGs along pseudotime between the two diverging lineages. We saw typical PT differentiation markers for lineage 1 such as *Slc34a1, Tmem27, Acsm2,* and *Slc27a2,* whereas cells at the

endpoint of lineage 2 demonstrated inflammatory and myeloid markers such as *Il1b, Cxcl2, S100a8,* and *S100a9* (Figs. 4e, S9f).

To unravel the gene regulatory network driving adaptive and maladaptive repair, we used *cis*-regulatory network analysis as implemented by SCENIC[39] and revealed transcription factor (TF)-centered gene co-expression networks ("Methods"). Binarized regulon activity ("on" or "off") clearly separated two trajectories (Figs. 4f, g, S9d and Supplementary Dataset 8). These two trajectories were consistent with our previous pseudotime analysis results. Clusters of repair lineage 1 (GMM1-3) showed specific enrichment e.g., of *Hoxa7, Nr1h4, Maf,* and *Hnf4a* regulons, while the maladaptive lineage 2 endpoint (GMM5) enriched for regulons *Cebpd, Fosl2, Atf3, Cebpb,* and *Ets2.* Gene expression of these representative lineage-differentiating TFs showed *Hnf4a* and *Maf* increase along lineage 1 and decrease along lineage 2, with vice versa relationships for fibrosis-driving TFs *Atf3, Cebpb, Cebpd, Ets2,* and *Fosl2* and corresponding predicted targets (Figs. 4h, j, S9f–h, j). Reassuringly, binarized regulon activity better highlighted TF enrichment specific to certain trajectory branches (Fig. 4i) than gene expression information alone (Fig. 4h). Maladaptive PT cells (in GMM cluster 5) showed the highest density of regulons among all PT cells (Fig. 4k, l).

**Pyroptosis and ferroptosis following sustained injury drive fibroinflammation**. To understand changes in control, injured, and maladaptive PT cells, we next identified DEGs for each time-point and ischemia dose (Supplementary Dataset 9). KEGG pathway analysis identified changes in proteasome pathways in all analyzed samples, as described before. Importantly, our data indicated early (day 1) changes in ferroptosis in both short and long IRI samples (Fig. 5a and Supplementary Dataset 10). Long IRI samples continued to show high levels of genes associated with ferroptosis (*Ptgs2, Chac1, Acsl4, Slc7a11,* and *Hmox1*), apoptosis (*Casp8, Bad, Bak1*), and pyroptosis (*Casp1, Casp4, Gsdmd, Il18,* and *Il1b*). Maladaptive PT cells showed marked pathway enrichment for ferroptosis and pyroptosis (caspase-1-mediated cell death in response to *Salmonella* or *Leishmania* spp.[40,41], Figs. 5a, b, S10a–c). At the same time, repairing PT cells showed enrichment for key homeostatic functions, such as oxidative phosphorylation and fatty acid degradation (Figs. 5a, S10d, e). This was also consistent with *Hnf4a, Maf,* and *Nr1h4* regulon activities (Figs. 4f–h, S9d). Reassuringly, we found some of the pyroptosis-associated transcripts from our IRI model such as *Aim2, Nlrp3, Pycard, Nod2, Naip2, Naip5* (inflammasome sensors and adapters), *Casp1, Casp4, Panx1, P2rx7, Casp8* (canonical and non-canonical activation of GSDMD), *Gsdmd* (pore forming executioner of pyroptosis), *Il18,* and *Il1b* (canonical downstream effectors) to be enriched in failed repair PT cells of a previous IRI dataset[34], in PT cells of a rejecting kidney allograft, and in PT cells of mice conditionally overexpressing Notch in tubules (Fig. S11a–c)[42]. We also corroborated evidence of increased GSDMD cleavage in IRI long samples compared to sham-treated samples in whole kidney lysates (Fig. S11d, e and Supplementary Dataset 11). In an orthogonal approach, we sought to validate pathway enrichment results by unbiased weighted gene coexpression network analysis (WGCNA, "Methods"). After generating metacells of the PT dataset (Fig. 6a, b), we retrieved 8 gene modules (Figs. 6c, S12a) demonstrating high specificity for IRI degree and timing post-ischemia (Figs. 6d, S12b). For example, the black module corresponded to PT cells 14 d after long ischemia (Figs. 6e, S12c and Supplementary Dataset 12). Pyroptosis markers *Gsdmd* and *Il1b* proved to be important hub genes (Fig. 6f) whose relatively high expression in maladaptive PT cells was highly specific (Fig. 6g, h). This was consistent with

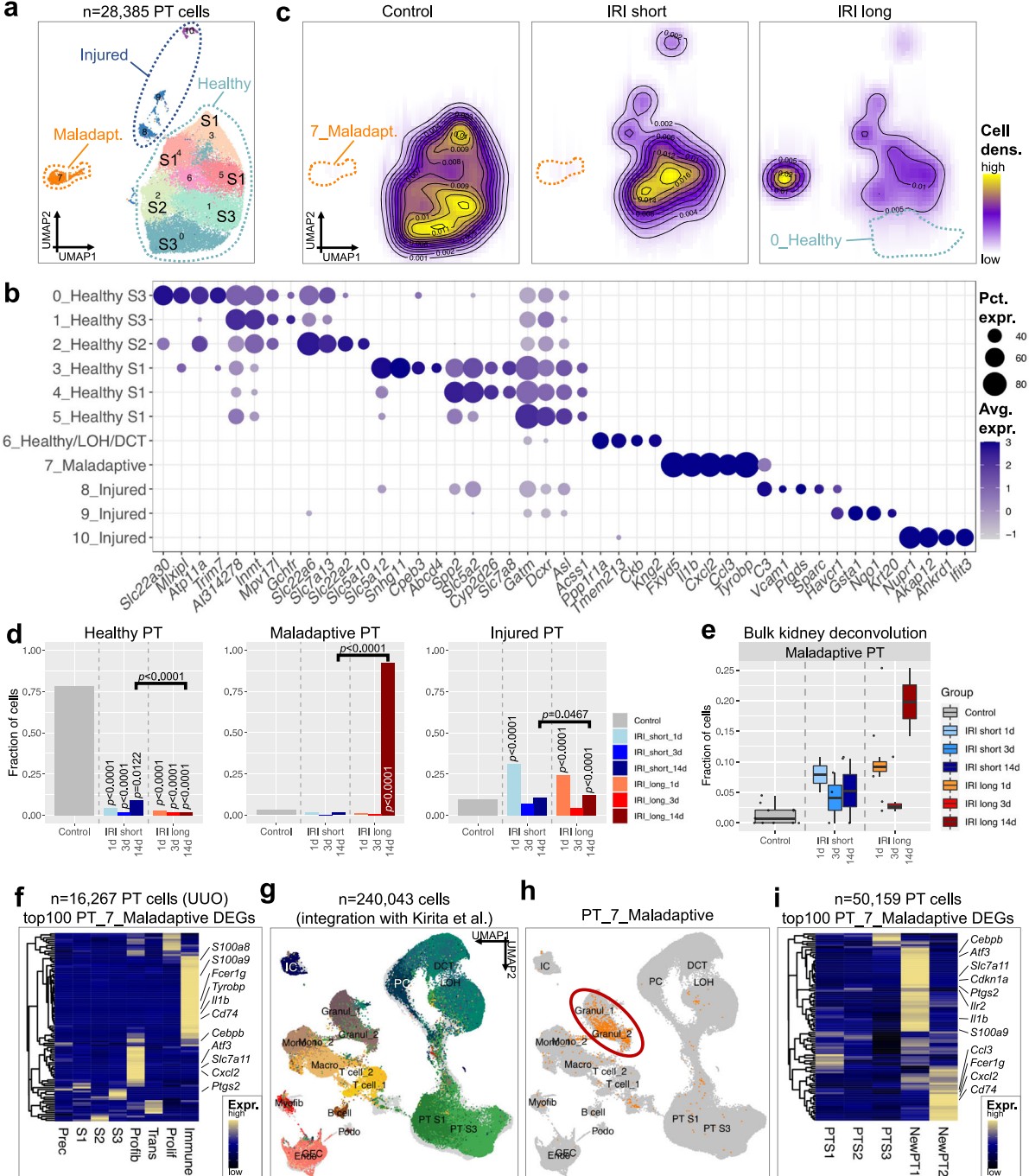

**Fig. 3 Profibrotic PT cells accumulate during maladaptive repair. a** UMAP projection of 28,385 PT cells, demonstrating 11 PT subclusters. Subclusters with a clear S1, S2, and S3 phenotype are annotated. **b** Corresponding dot plot showing PT subcluster-specific expression patterns of differentially expressed genes (DEGs). **c** Changes in cell type proportions of PT cells between different treatment groups (Control; IRI short; IRI long), visualized as cell density. Control and IRI short samples are dominated by healthy PT subclusters, IRI long samples by Maladaptive PT. **d** Fractions of Healthy, Maladaptive, and Injured PT cells stratified by ischemia dose and time post-IRI. Statistical significance for comparisons in $n = 28,385$ cells was derived using differential proportion analysis, with a mean error of 0.1 over 100,000 iterations; $p$ values are given for comparisons between Control and IRI groups and between short and long IRI groups, respectively. **e** Fraction of cells annotated as Maladaptive PT after bulk kidney RNA-seq data cell type deconvolution using BisqueRNA, displayed as Tukey box plots. $X$-axis denotes the different treatment groups (Control, IRI short 1 d, IRI short 3 d, IRI short 14 d, IRI long 1 d, IRI long 3 d, IRI long 14 d). Corresponding results for the MuSiC deconvolution pipeline are shown in Fig. S8a. **f** Heatmap showing the expression of the top 100 differentially expressed genes (DEGs) from PT_7_Maladaptive in 16,267 PT cells from fibrotic mouse kidneys undergoing unilateral ureteric obstruction (UUO) highlights significant overlap with profibrotic (Profib) and immune phenotype clusters. Top DEGs are highlighted on the right. **g** Joint UMAP embedding after integration of all 113,579 kidney cells with the dataset from Kirita et al. (126,464 kidney cells). Labels and colors represent clustering as in Fig. 1f. Cells from Kirita et al. are labeled in gray. **h** Cells annotated as PT_7_Maladaptive are colored orange in the joint UMAP embedding. Note similar projection within the joint embedding space. Cluster labels as in Fig. 1f. **i** Heatmap showing the expression of the top 100 differentially expressed genes (DEGs) from PT_7_Maladaptive in 50,159 PT cells from Kirita et al. highlights significant overlap with injured clusters "NewPT1" and "NewPT2". Top DEGs are highlighted on the right.

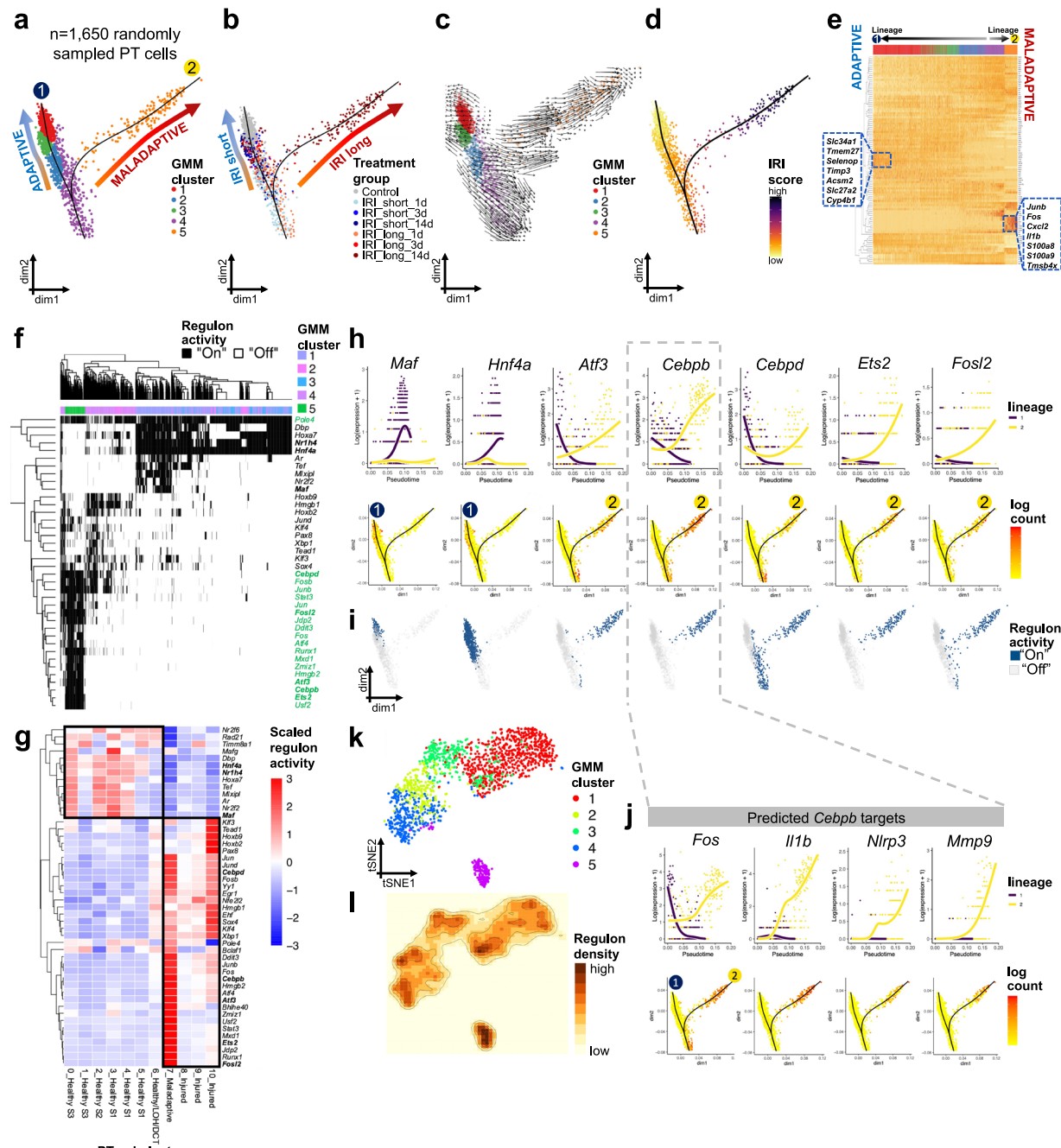

**Fig. 4 PT trajectories of adaptive and maladaptive repair. a, b** Slingshot-derived trajectories of 1,650 randomly sampled PT cells using Gaussian mixture modeling (GMM) yielded 5 clusters in 2 trajectories. GMM clusters (**a**) and treatment groups (**b**) demonstrated major overlap: Trajectory 1 represented adaptive repair of acutely injured cells 1 d post-ischemia (GMM4) towards endpoint clusters (GMM1&3) corresponding to control and short IRI 14 d samples. Trajectory 2 represented maladaptive repair of acutely injured cells (GMM4) towards fibroinflammation (GMM5). **c** RNA velocity analysis in the same 1650 PT cells recapitulated similar trajectories. Cells are colored by GMM clusters as in (**a**). **d** Steady IRI score increase along trajectory 2. **e** Heatmap showing generalized additive modeling (GAM)-derived differentially expressed genes (DEGs) along the 2 PT lineages following IRI. Rows represent DEGs, columns represent individual PT cells. Color legend at the top corresponds to GMM clusters from (**a**). The lineage 1 endpoint was exclusively from samples 14 d after short IRI and Controls, demonstrating increased expression of PT differentiation markers (e.g., *Slc34a1, Acsm2*). The lineage 2 endpoint was exclusively from samples 14 d after long IRI and showed increased inflammatory (e.g., *Il1b, Cxcl2, S100a8,* and *S100a9*) and stress marker (e.g., *Junb, Fos*) expression. **f** Heatmap of cell type-specific binarized regulon activity, as inferred by *cis*-regulatory network analysis using SCENIC. Rows represent regulon activity (binarized to "on" = black or "off" = white), columns represent individual PT cells. Color legend at the top displays GMM clusters. **g** Heatmap of PT subcluster-specific scaled regulon activity. Rows represent regulons, columns represent PT subclusters. **h, i** Gene expression and regulon activity of exemplary lineage-specific transcription factors (TFs) governing top specific regulons. **h** Pseudotime-dependent gene expression along the 2 PT lineages and corresponding feature plots. **i** Binarized regulon activity (blue = "on", gray = "off"). Regulon activity and gene expression largely overlapped and demonstrated lineage-specific patterns. **j** Gene expression of *Cebpb* targets, as predicted by motif analysis, showed strong increases of proinflammatory genes such as *Il1b* and *Nlrp3* along lineage 2. **k** tSNE reduction plot of GMM clusters showing good separation of trajectory-inferred clustering. **l** tSNE representation of regulon density as a surrogate for stability of regulon states.

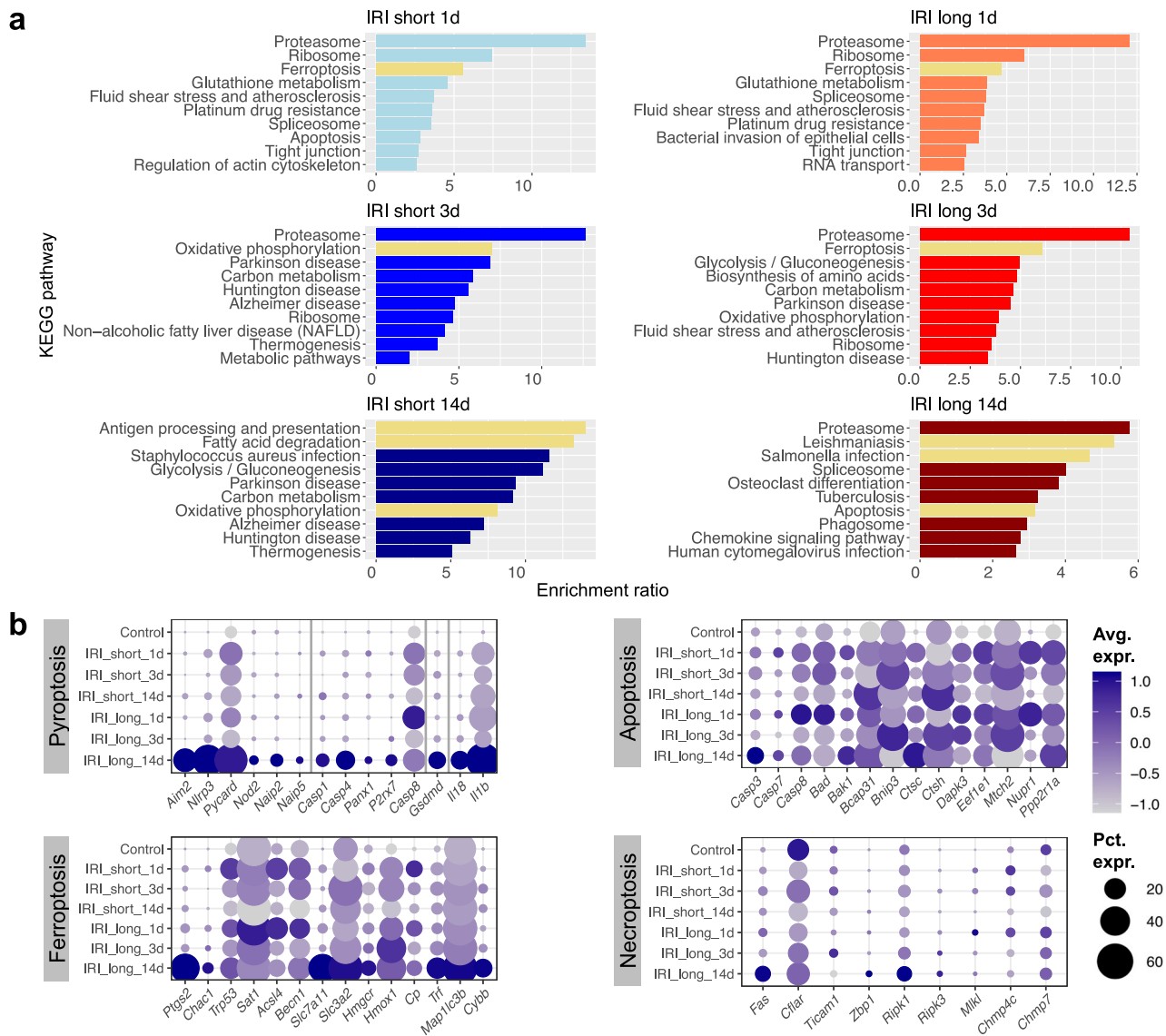

**Fig. 5 Increase in pyroptosis and ferroptosis following sustained injury drives inflammation. a** Enrichment ratios of top enriched KEGG pathway terms, derived by overrepresentation analysis of DEGs in proximal tubule cells. The graph is split into subpanels by IRI degree (short and long, respectively) as well as by time point post ischemia (1, 3, and 14 d). **b** Dot plots showing experimental group-specific expression of genes in PT cells involved in regulated cell death pathways pyroptosis, ferroptosis, apoptosis, and necroptosis. Dot size denotes percentage of cells expressing the marker. Color scale represents average gene expression values.

KEGG pathway analysis (Fig. S13 and Supplementary Data-set 13). Finally, we validated gasdermin D as pyroptosis marker with in situ hybridization (Fig. 6i) and immunofluorescence (Fig. 6j), demonstrating indeed presence of *Gsdmd* mRNA and GSDMD protein in PT cells of maladapted kidneys.

**Cell–cell interaction landscape after short and long ischemia.** Next, we sought to investigate the complex network of cell–cell communication following severe and moderate IRI ("Methods"). Control kidneys demonstrated only limited interactions between immune (mostly T) cells and epithelial, endothelial, or stromal cells (Fig. S14a). More cell–cell interactions were observed after IRI and even greater after long ischemia (Fig. S14a–c). Top signals of ligand–receptor interaction involving PT cells included typical immune signaling such as ligand *Cxcl2* (expressed by maladaptive PT cells and myeloid cells) to its receptor *Cxcr2* (expressed by granulocytes). *Il1b* and its receptors *Il1r2* and

*Adrb2* were expressed not only by myeloid cells but also by maladaptive PT cells. *Ccl3* and its receptor *Ccr1* and *Il34* and its receptor *Csf1r* showed similar patterns (Fig. 7a). The maladaptive PT cluster showed high expression of immune cytokines and typical myeloid markers, which we validated in an external dataset[43] (Figs. 7a, S15a–d). In fibrotic kidneys, inflammatory signaling connections were active in maladaptive PT cells and myeloid cells (PT-to-immune) as well as among epithelial cells (maladaptive PT-to-maladaptive PT) (Fig. 7a, b). For example, while IL1B signaling was active within the immune compartment at baseline, in long IRI and fibrosis maladaptive PT cells became a signaling node. Finally, ligand–receptor interactions between myeloid cells and profibrotic PT increased substantially following long IRI (Fig. 7b), highlighting the role of profibrotic PT in attracting myeloid cells.

Intrigued by these results and the high enrichment of immune cells in our dataset, we then set out to characterize the immune cell landscape of the kidney following short and long ischemia.

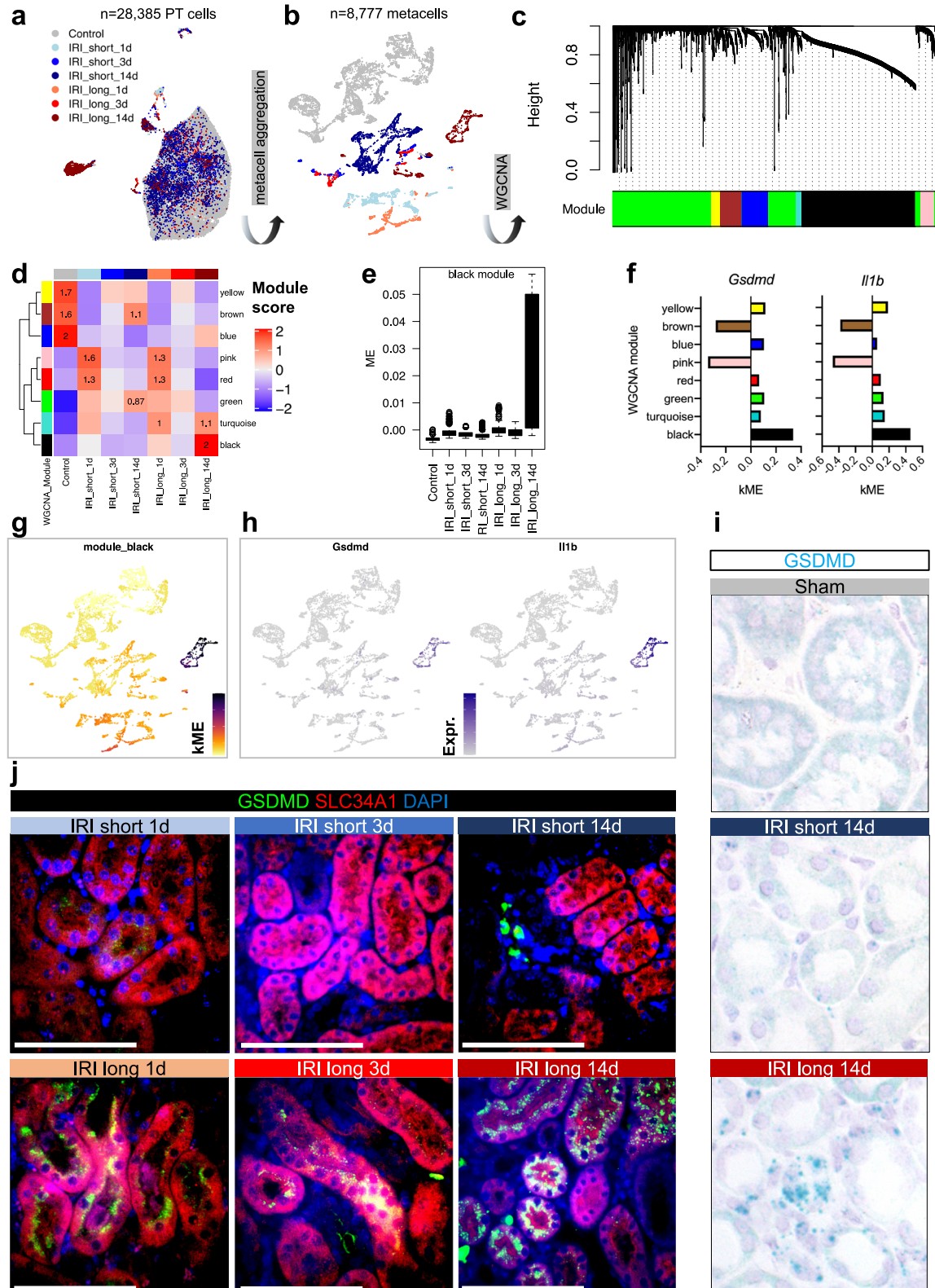

**Fig. 6 WGCNA, in situ hybridization, and immunofluorescence confirm GSDMD as maladaptative repair-specific hub gene expressed in PT. a** UMAP visualizing PT cells colored by IRI degree and time post-ischemia. **b** UMAP visualizing PT metacells colored as in (**a**). **c** Hierarchical cluster tree showing gene co-expression modules identified by weighted correlation network analysis (WGCNA) in PT cells revealed 8 modules (color-coded). **d** Heatmap demonstrating high WGCNA module specificity among treatment groups stratified by IRI degree and time post-ischemia. **e** Module eigengene (ME) scores of black WGCNA module in $n = 8777$ metacells by IRI degree and time post-ischemia, displayed as Tukey box plots (outliers denoted as dots outside box plot whiskers). **f** Intramodular connectivity (kME) values for individual hub genes *Gsdmd* and *Il1b* show high specificity for black module. **g**, **h** kME values (**g**) and *Gsdmd* and *Il1b* expression (**h**) visualized in UMAP space. **i** In situ hybridization showing presence of *Gsdmd* mRNA in IRI long 14 d kidney tubules. **j** Immunofluorescence staining representative of $n = 3$ independent experiments of GSDMD (green) and PT marker SCL34A1 (red); scale bars = 50 µm.

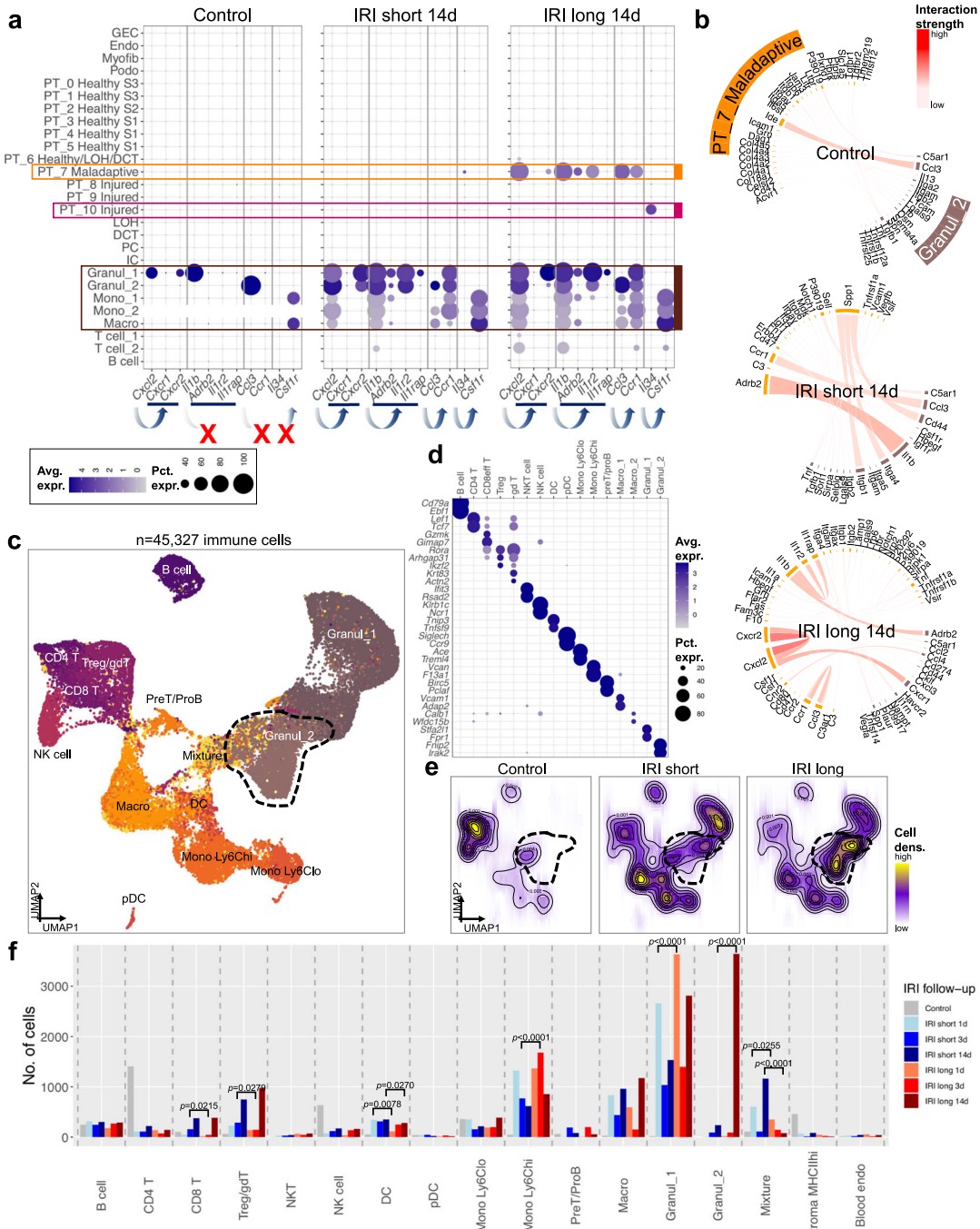

**Fig. 7 Transcriptional atlas of immune cells in adaptive and maladaptive kidneys. a** Dot plots showing an increase of expression of genes corresponding to proinflammatory cell–cell interactions among myeloid cells after IRI. 14 d after long IRI, maladaptive PT adopts an immune cell phenotype. Dot size denotes percentage of cells expressing the marker. Color scale represents average gene expression values. Arrows indicate ligand–receptor pairs, as derived from CellPhoneDB. X denotes abrogated signaling. **b** Circos plots visualizing genes involved in ligand–receptor interaction between PT_7_Maladaptive and corresponding interaction partner cells (Granul_2). Color scale is proportional to interaction strength, as measured by CellPhoneDB interaction means of statistically significant interactions. Note that signaling is both between epithelium and myeloid cells as well as exclusively among epithelium. **c** UMAP projection of 45,327 immune cells. Clusters encompassed B cells, CD4 and CD8 T cells, regulatory and γδ T cells (Treg/gdT), natural killer T (NKT), natural killer (NK) cells, dendritic cells (DC), plasmacytoid DC (pDC), monocytes with high and low expression of Ly6C, respectively (Mono Ly6C^lo, Mono Ly6C^hi), PreT/proB precursor cells, macrophages (Macro), Granulocytes (Granul_1, Granul_2), a mixed phenotype cluster (Mixture), MHCII expressing stroma cells (Stroma MHCII^hi), and blood endothelial cells (Blood endo). **d** Dot plot showing cell type-specific expression patterns of marker genes. Dot size denotes percentage of cells expressing the marker. Color scale represents average gene expression values. **e** Changes in cell proportions of immune cell types between treatment groups are visualized as cell density. Cells from Control samples were most dense in T cell clusters, those from IRI short samples in Macro, Mono Ly6C^hi, and Granul_1 clusters, and those from IRI long samples in the Granul_2 cluster. **f** Bar graph depicting the number of different immune cells (*n* = 45,327 total cells) according to treatment group (Control; IRI short; IRI long) and according to time point after ischemia (13, and 14 d). Statistical significance for comparisons between short and long IRI was derived using differential proportion analysis, with a mean error of 0.1 over 100,000 iterations; *p* values are given for time point comparisons between IRI short and long.

Subclustering revealed intricate diversity of cell types (Figs. 7c, d, S16a) with specific enrichment of CD4⁺ T and NK cells in control, CD8⁺, Treg/γδT, and dendritic cells in adaptively repaired (IRI short), and Ly6Cʰⁱ monocytes, and granulocytes in maladapted (IRI long) kidneys, respectively (Figs. 7e, f, S16b). A subset of granulocytes (Granul_2) was highly specific to maladapted kidneys (Figs. 7f, S16c) and showed strong enrichment of cell death pathways by GO term analysis (Fig. S16d–f).

In summary, long ischemia led to sustained enrichment of pyroptosis and ferroptosis in PT cells. Profibrotic PT cells contributed the most to these changes and were likely responsible for attracting myeloid cells that were enriched in fibrotic kidneys.

**Druggability screen identifies pyroptosis and ferroptosis as key driver pathways for maladaptive repair.** To gain further insight into pathways driving the maladaptive kidney response and to identify potential candidates for therapeutic interventions, we queried the LINCS database of drug response patterns for overlap with our observed maladaptive kidney signature. Drug prototype ranked lists (PRLs) were calculated for L1000 drug responses as described recently[44] and queried for DEGs specific to maladaptive PT cells using GSEA (Fig. 8a, "Methods")[45]. Top drugs with positive normalized enrichment scores (NES) for the maladaptive signature included crizotinib and erlotinib (Figs. 8b, S17a, b). These drugs have been shown to prevent fibrosis development in experimental models of kidney disease although both drugs are known to induce pyroptosis[46,47], which we confirmed by quantitative real-time PCR (qPCR) in vitro analyzing primary mouse renal tubular epithelial cells (Fig. S18a–d, Supplementary Dataset 14); we also confirmed erlotinib and crizotinib did not induce ferroptosis via qPCR and live cell imaging of lipid peroxidation (Fig. S18d, e). Furthermore, leading edge analysis of the top 24 enriched drugs confirmed the strongest drug-induced upregulation to affect IL1B, FCER1G, TYROBP, and CXCL2 (Fig. 8c), suggesting involvement of pyroptosis.

Having found pyroptosis and ferroptosis as candidate druggable pathways in the mouse single-cell data, we analyzed human CKD samples with fibrosis. We examined microdissected human kidney tubule RNA-seq profiles of $n = 433$ human kidney biopsy samples[48]. Expression of genes associated with pyroptosis (CASP1, NLRP3) and ferroptosis (ACSL4, CYBB) showed strong and significant positive correlation with fibrosis ($R = 0.66, 0.51, 0.40$, and $0.55$) and negative with eGFR ($R = -0.41, -0.26, -0.34$, and $-0.34$) (Fig. 8d). Similar relationships were seen for genes specific to maladaptive PT injury (Fig. S19a), while relationships were inverse for successful PT repair genes (Fig. S19b).

In summary, our single-cell analysis highlighted pyroptosis and ferroptosis in maladaptive PT cells and potential druggable pathways in maladaptive PT repair.

**Pharmacological inhibition of pyroptosis and ferroptosis ameliorates maladaptive kidney response and fibrosis after severe IRI.** Finally, we sought to confirm our computational results by using pharmacological inhibitors of pyroptosis and ferroptosis or vehicle in our severe IRI mouse model (Fig. 9a). We repeated the long IRI model and treated mice with VX-765 (pyroptosis inhibitor) and liproxstatin (ferroptosis inhibitor) for 14 d, we then sacrificed animals and repeated the single-cell analysis. Integration of inhibitor-treated kidney scRNA-seq samples with our previous dataset of short and long IRI yielded 133,433 kidney cells after quality control (Figs. 9b, S20a).

We observed preserved PT cell numbers in the inhibitor-treated animals (Fig. 9c). Consistent with the notion that maladaptive PT cells attract immune cells via cytokine secretion, we observed lower myeloid and immune cell fractions in mice

treated with pyroptosis and ferroptosis inhibitors (Fig. 9d, e). Cell fractions and densities of inhibitor-treated kidneys resembled control kidneys (Figs. 9c–e, S20b, c) and myeloid and maladaptive/profibrotic PT cells-driven high IRI scores in fibrotic kidneys were reduced to control levels by pharmacological inhibition (Fig. 9f, g).

We then aimed to dissect putative driver TFs of the observed protective drug response by employing trajectory and motif enrichment analyses in all 47,791 PT cells of this integrated dataset. We found two trajectories towards adaptive (lineage 1) and maladaptive repair (lineage 2) (Fig. 9h). Intriguingly, both lineages were connected by inhibitor-treated cells (Fig. S20d, e) demonstrating a low IRI score (Fig. S20f). Motif enrichment analysis confirmed the highest regulon density in this cluster (Figs. S20g, S21a) and a number of distinctly activated regulons for clusters along lineages 1 and 2, respectively (Fig. S21b, c): Lineage 1 (adaptive) showed specific enrichment of regulons such as *Esrra, Foxo3, Hnf4a, Maf, Pax2, Ppara, Ppargc1a,* and *Tead1,* important for oxidative phosphorylation, fatty acid oxidation, and PT differentiation. Lineage 2 (maladaptive) enriched for regulons *Jun, Junb, Jund, Fos, Fosb, Atf3, Atf4, Stat3, Stat5a Irf2, Irf5, Irf8, Irf9, Runx1, Runx3, Egr1, Cebpa, Cebpb, Cebpd* (Fig. S21c), important for stress, injury, and proinflammatory response as well as myeloid differentiation.

Finally, following pharmacological inhibition of pyroptosis and ferroptosis, reduced gene expression of important marker genes for these pathways (Fig. 9i) resulted in amelioration of functional and structural AKI (Fig. 9j–l and Supplementary Dataset 15). BUN and creatinine were significantly ameliorated in VX-765 vs. vehicle-treated mice at all timepoints up to 14 d post-ischemia, whereas liproxstatin only resulted in structural but not functional amelioration. However, irrespective of kidney function, both VX-765 and liproxstatin treatment resulted in significant reduction of fibrosis 14 d post-IRI despite severe ischemia (Fig. 9k–l).

**Discussion**

Here we present a comprehensive temporal and cell type resolution analysis of adaptive and maladaptive kidney repair. We identify key cell type-specific regulatory networks driving both renal adaptive repair and maladaptive/fibrotic repair during the AKI-to-CKD transition. We show that in PT cells, which are most susceptible to toxic and hypoxic injury[48], necrotic cell death pathways such as pyroptosis and ferroptosis demonstrate sustained enrichment and are associated with maladaptive AKI-to-CKD progression. Finally, leveraging drug response data, we confirm in vivo that targeting of pyroptosis and ferroptosis ameliorated maladaptive injury signatures despite severe IRI, highlighting the druggability of necrotic cell death pathways.

A variety of cell death pathways has been described during AKI, including apoptosis and regulated necrosis (ferroptosis, pyroptosis, and necroptosis)[49–51]. The data on their role in injury and repair is controversial[52,53]. For example, while inhibitor-based studies indicated a key role of apoptosis in AKI[54,55], it can also be a beneficial, non-inflammatory form of cell death[53]. A key bottleneck in determining the role of cell death pathways has been that prior methods have been unable to generate cell type-specific read-outs. Our comprehensive single-cell resolution analysis of adaptive and maladaptive IRI fills in a key information gap. Consistent with previous studies highlighting the importance of ferroptosis for causing synchronized regulated necrosis in renal tubules early after AKI[56–60], we found early changes in ferroptosis in both short and long AKI models. We observed important differences, as ferroptosis and pyroptosis were markedly pronounced in the long AKI model. We found genes associated with pyroptosis, such as *Casp1, Casp3, Casp4, Il1b, Il18,* and *Gsdmd,* as

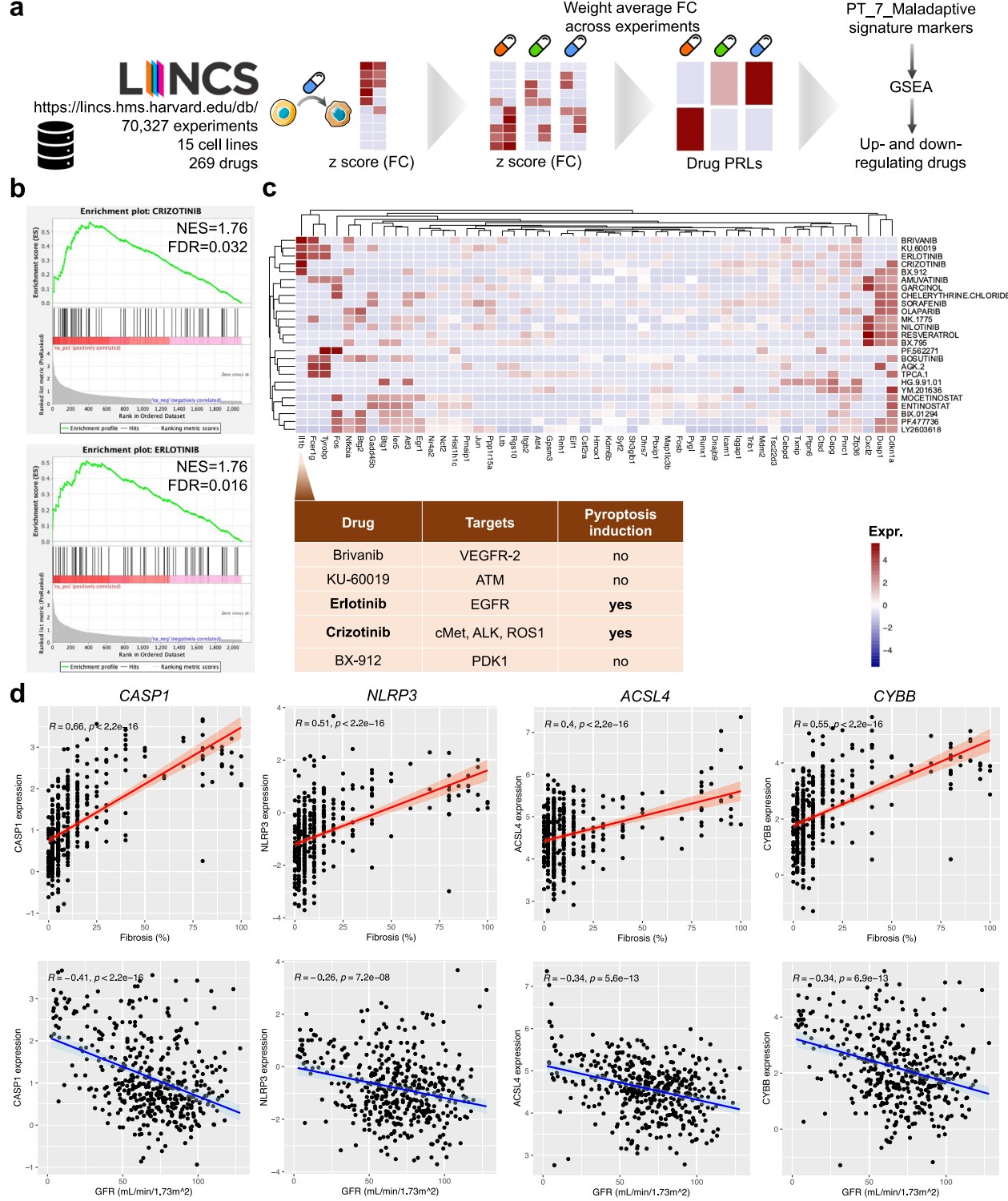

**Fig. 8 Druggability screen identifies pyroptosis and ferroptosis as key driver pathways for maladaptive repair. a** Experimental setup: Identifying candidate molecules for drug: Gene expression values were calculated from experiments in the LINCS database and used to select drugs that would up- or downregulate the genes identified as Maladaptive PT markers. **b** Gene set enrichment analysis (GSEA) plots of top 2 drug response patterns overlapping with pre-ranked maladaptive DEGs ("Methods"). **c** Corresponding leading edge analysis heatmap visualizing expression of core genes enriched in the respective top 24 drug response patterns overlapping with pre-ranked maladaptive DEGs. *Il1b* was among the genes upregulated most frequently and the drugs eliciting Il1b upregulation are listed in the corresponding table along with their target and whether they induce pyroptosis or not. **d** Correlation of pyroptosis (*CASP1, NLRP3*) and ferroptosis (*ACSL4, CYBB*) marker gene expression in microdissected human kidney tubules with fibrosis (top row) and eGFR (bottom row); error bands represent 95% confidence interval.

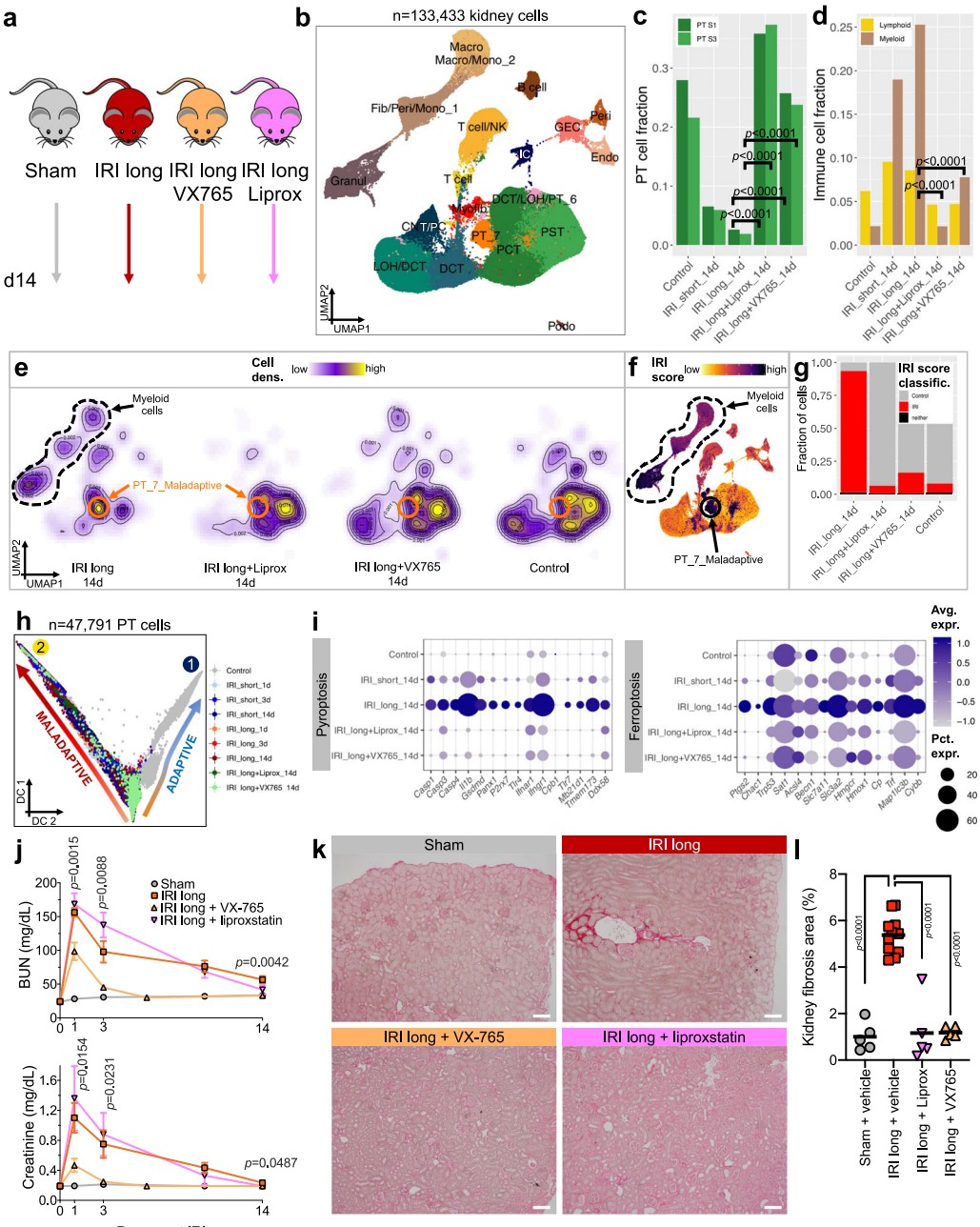

**Fig. 9 Pharmacological inhibition of pyroptosis and ferroptosis ameliorates maladaptive kidney response and fibrosis after severe IRI. a** Experimental setup. Male C57BL/6 mice were treated 30 min before and daily after long (30 min) bilateral renal ischemia with vehicle (IRI long), VX-765 (pyroptosis inhibition), or Liproxstatin (ferroptosis inhibition) for 14 d. Sham-operated animal served as controls; artwork own production and from https://smart.servier. com, license https://creativecommons.org/licenses/by-sa/3.0/). **b** UMAP projection of 133,433 cells of $n = 6$ controls, $n = 6$ IRI short, $n = 6$ IRI long, $n = 1$ IRI long+VX765, and $n = 1$ IRI long+Liprox samples passing rigid quality control filtering (nFeatures > 200 and < 3000, mt% < 50, doublet removal) and after dataset integration with LIGER. **c**, **d** Fractions of PT (**c**) and immune cell (**d**) clusters stratified by treatment. Statistical significance for comparisons was derived using differential proportion analysis, with a mean error of 0.1 over 100,000 iterations; $p$ values are given for comparisons between inhibitor and vehicle-treated IRI long groups. **e** Highest cell densities in IRI long samples were among PT_7_Maladaptive and myeloid clusters, whereas inhibitor-treated and control samples showed highest cell densities in healthy PT clusters. **f** IRI scores were highest in myeloid and PT_7_Maladaptive clusters. **g** Fractions of IRI-classified cells were drastically reduced in inhibitor-treated and control samples. **h** Diffusion map dimensional reduction of Slingshot-derived cell trajectories of 47,791 PT cells using Gaussian mixture modeling (GMM) yielded 9 clusters in 2 trajectories. Trajectory 1 encompassed mostly IRI short and control samples, trajectory 2 encompassed long IRI samples. Inhibitor-treated samples were right at the junction between lineages 1 and 2. **i** Dot plots showing experimental group-specific expression of genes in PT cells involved in regulated cell death pathways pyroptosis and ferroptosis. Dot size denotes percentage of cells expressing the marker. Color scale represents average gene expression values. **j** Functional effects of pyroptosis and ferroptosis inhibition on long IRI. Blood urea nitrogen (BUN) and creatinine changes (right panel) 1, 3, and 14 d post-IRI; means ± SEM; $p$ values are given for multiple t tests (Benjamini, Krieger, Yekuteli corrected) comparing IRI + VX-765 ($n = 5$) vs. IRI + vehicle ($n = 11$). IRI + Liprox ($n = 5$) vs. IRI + vehicle ($n = 11$) comparisons were non-significant; $n = 3$ independent experiments, $n = 5$ Sham samples. **k** Representative light microscopy of Sirius red-stained kidneys; scale bars = 50 μm. **l** Fibrosis quantification from (**k**); $p$ values are given for one-way ANOVA (Holm–Sidak corrected).

well as markers for ferroptosis, such as *Ptgs2, Chac1, Acsl4, Slc7a11,* and *Hmox1,* to be highly enriched in PT cells from samples following long IRI compared to all other samples. Interestingly, apoptosis markers such as *Casp8, Bad, Bak1, Nupr1,* and *Ppp2r1a* were similarly enriched in IRI samples irrespective of the damage degree, when compared to controls. It is important to note that necroptosis marker enrichment was not particularly strong in any of the PT cells, indicating that necroptosis might be important in non-PT cells[61]. Our observations are in keeping with a growing body of evidence that pyroptosis and the NLRP3 inflammasome are intertwined with AKI and fibrogenesis. For example, it has been shown that pyroptosis was upregulated in cisplatin[62,63] and IRI models[62,64,65] and *Gsdmd* and *Casp11*-deficient mice were protected against cisplatin-induced AKI[62]. Others demonstrated that NLRP3 facilitated TGFβ/Smad signaling[66], while *Nlrp3*-deficient mice were protected against diabetic nephropathy and renal fibrosis[67], highlighting pyroptosis as a critical pathway promoting both AKI and profibrotic transition to CKD.

In our analysis, inflammatory cell death was a key nidus for ensuing inflammation. Maladaptive cells expressed several inflammatory and immune TFs and a variety of cytokines, such as *Il1b, Il34,* and *Cxcl2* etc. Cell–cell interaction analysis indicated that these cytokines play key roles in attracting myeloid cells and fibroblasts via the maladaptive PT cluster. Indeed, we observed major changes in immune, specifically myeloid cell fractions in fibrotic samples. It is important to note that these cytokines signaled to not only immune cells but other PT cells, likely maintaining the vicious cycle of maladaptive repair. Myeloid cells are known to play a key role in fibrosis development and were shown to be highest 28 d post-IRI in a previous study[68,69].

Our results indicate that single-cell transcriptomics are not only well-suited to identify changes in cell fraction and cell type-specific gene expression but can be effectively used for identifying druggable pathways for intervention. Our analysis highlighted ferroptosis and pyroptosis as key targetable pathways. Indeed, treatment of mice with VX765 or liproxstatin protected from maladaptive repair and fibrosis. It was interesting to note that while VX765 prevented the acute rise in BUN and creatinine, the effect of liproxstatin was more pronounced on preventing long-term structural damage and fibrosis development. These results indicate that protecting from CKD and fibrosis is more complex than just looking at biochemical markers of kidney function[70].

An important limitation of our work is that we could not fully distinguish the targets of VX765 and liproxstatin at the single-cell level and we cannot fully exclude that some of their impact was not due to off-target effects, given that interdependencies of the several necrotic cell death forms have already been demonstrated[60]. Future studies will be needed to elucidate the individual contributions and potentially intricate interdependencies of necrotic cell death pathways. Our in silico drug database screen used in vivo experimental data rather than just gene or pathway predictions. Future studies shall examine these pathways in a more detailed manner.

Taken together, we demonstrate that maladaptive repair following injury is characterized by an inflammatory profibrotic PT cell phenotype that secretes large amounts of cytokines and is responsible for myeloid cell influx and fibrosis. We show that single-cell analysis can identify druggable pathways for intervention. Finally, we demonstrate that in maladaptive PT cells ferroptosis and pyroptosis drive immune cell activation and fibrosis development.

## Methods

**Renal ischemic reperfusion injury model.** Male C57Bl/6J mice were obtained from Jackson Laboratories (Bar Harbor, ME, stock no. 000664). Animal studies (protocol #804138) were approved by the Institutional Animal Care and Use Committee (IACUC) of the University of Pennsylvania. Mice were housed in a pathogen free environment (12 h dark/light cycle) and fed with standard mouse diet and water ad libitum. All surgeries were performed by the same investigators who were blinded to the experimental grouping by using a de-identifying numbering scheme. 10–12 wks old mice were anesthetized with isoflurane and placed on an automatic closed loop temperature control system (Homeothermic Monitoring System, Harvard Apparatus, Holliston, CA). Core body temperature as measured via rectal probe was maintained at 36.8–37.2 °C throughout the procedure. Kidneys were exposed via dorsal incision. Renal pedicles of both kidneys were exposed and bilateral clamping was performed with arterial clips to induce bilateral renal ischemia, which was visually confirmed by color change. Six animals were subjected to moderate renal ischemia (23 min), whereas another 6 mice were subjected to severe injury (30 min). Buprenorphine SR (0.1 mg/kg s.c.) was administered after release of arterial clips and before weaning of anesthesia. Animals were sacrificed and blood and kidneys were sampled 1, 3, and 14 d post-ischemia, respectively.

In a second set of experiments, mice were treated with intraperitoneal injection of inhibitors or vehicle 30 min before as well as every 24 h for 14 d after long IRI (30 min). In a sham group, renal pedicles were not clamped after bilateral visualization, and anesthesia was maintained for the identical time. Inhibitors VX-765 (100 mg/kg) and liproxstatin (10 mg/kg) were dissolved in DMSO and formulated in sesame oil to final concentrations of 16.67 and 1.67 mg/mL, respectively, as in previous studies[56,71]. The vehicle group received equal amounts of DMSO.

**Blood creatinine and blood urea nitrogen.** To estimate kidney function, blood creatinine and BUN were measured at time points as indicated with VetScan iSTAT CHEM8 + cartridges (Abaxis, Union City, CA).

**Histopathological and immunofluorescence analysis.** Kidneys samples were fixed in 10% neutral formalin and paraffin-embedded sections were stained Periodic acid-Schiff (PAS) and Sirius red. Samples were scored by an experienced pathologist in a blinded fashion for signs of acute tubular injury. Fibrosis was quantitated in Sirius red-stained sections using MRI fibrosis tool and Color Deconvolution plugin for "ImageJ [https://github.com/MontpellierRessourcesImagerie/imagej_macros_and_scripts/wiki/MRI_Fibrosis_Tool]". Ten similarly oriented photos in the same part of the kidney at the same magnification were studied per biological sample. Color deconvolution and simple auto-thresholding were applied and the fibrosed area of the selection was measured and compared to the area of the selection on the input image.

Immunofluorescence staining was performed after deparaffinization and antigen retrieval using citrate buffer pH 6.0 with a pressure cooker (PickCell Laboratories, Agoura Hills, CA). Antibodies were diluted in blocking buffer (TBS 0.1% Tween 20, 0.05% Triton X-100, 5% bovine serum albumin) as follows: anti-GSDMD (1:100, Santa Cruz, #393656), anti-SLC34A1 (1:100, Novus, #NBP2-13328), anti-CD11B (1:50, BD, #555386), AF555-conjugated donkey anti-rabbit (1:1000, LifeSciences, #A31572), AF488-conjugated goat anti-mouse (1:1000, LifeSciences, #A11029).

**Flow cytometry.** Mice were perfused with 10 mL PBS prior to organ procurement. Kidneys were diced and digested in a solution of 1 mg/mL collagenase A (Roche) and 100 mg/mL DNase (Roche) in complete RPMI for 1 h at 37 °C to obtain a single cell suspension. Cells were homogenized by passing through an 18G needle 5 times and filtered through a 100 µm mesh. Cells were then washed with FACS buffer (1× PBS, 2.5% FBS, 2 mM EDTA) before incubating with Fc block (99.5% FACS buffer, 0.5% normal rat serum, 1 µg/mL 2.4G2 IgG antibody) prior to staining. Cells were stained with the viability dye Ghost Dye Violet 510 (Tonbo biosciences, #12-0870) and antibodies used for staining are shown in the Supplementary Table 1. Samples were run on FACSymphony A3 (BD Biosciences) and analyzed using the FlowJo Software analysis program (TreeStar). Gating strategy is shown in Fig. S6.

**In situ hybridization.** In situ hybridization was performed using formalin-fixed paraffin-embedded tissue samples and the RNAscope® 2.5 HD Duplex Detection Kit (ACD #322436) with Mm-*Gsdmd* probe #537601. We followed the manufacturer's original protocol.

**Isolation and culture of primary mouse tubular epithelial cells.** Primary mouse tubular epithelial cells were isolated from kidneys of 2–4 wks old C57Bl/6J mice (Jackson Laboratories, Bar Harbor, ME, stock no. 000664). Cells were grown in RPMI 1640 supplemented with 10% fetal bovine serum (FBS), 20 ng/mL epithelial growth factor (Peprotech #AF-100-15), 1× ITS (Gibco #51500-056), and 1% penicillin-streptomycin (Corning #30-002-CI) at 5% $CO_2$ and 37 °C. Prior to drug treatment, cells were grown for 12 h in media supplemented with 1% FBS. Cells were then treated for 24 h with different doses of erlotinib, crizotinib, and erastin (all Cayman Chemical), respectively. Dimethyl sulfoxide (DMSO, Merck) was used as vehicle control. Lactate dehydrogenase (LDH) release was measured using CytoTox 96 Non-Radioactive cytotoxicity assay (Promega), cytotoxicity and

viability were measured by using MultiTox-Fluor Multiplex cytotoxicity assay (Promega) as per the manufacturer's instructions.

**Live cell imaging**. Primary cultured renal tubule cells were incubated with 1.5 μM BODIPY™ 581/591 C11 (Thermo Fisher #D3861) for 30 min at 5% $CO_2$ and 37 °C. After staining with Hoechst 33342 at a final concentration of 1 μg/mL in PBS, cells were imaged directly with a Keyence fluorescence microscope.

**qPCR**. RNA was isolated from cells using Trizol (Invitrogen). 2 μg RNA was reverse-transcribed using cDNA Reverse Transcription Kit (Applied Biosystems, #4368813), and qPCR was run on a ViiA 7 System (Life Technologies) using SYBR Green Master Mix (Applied Biosystems, #4367659) and gene-specific primers. For quantitative analysis, samples were normalized to glyceraldehyde-3-phosphate dehydrogenase (*Gapdh*) with the ΔΔCt method. Primer sequences are listed in Supplementary Dataset 16.

**Western blot**. Nearly 15 mg of total kidney tissue was homogenized in SDS lysis buffer (Cell Signaling Technology, CST, #7722), sonicated, and heated at 95 °C. Lysates were cleared by centrifuging (14,000 × g at 4 °C for 10 min). 10 μL of total lysate was loaded onto 11% SDS-PAGE and subjected to electrophoresis (100 V, room temperature). Proteins were transferred onto PVDF membranes at 100 V on ice for 1 h. Membranes were incubated in 3% non-fat skimmed milk solution prepared in Tris-buffered saline containing Tween-20 (TBST) for 1 h at room temperature on an orbital rocker. Membranes were probed with anti-GSDMD (Ab209845, abcam, dilution 1:1000) and anti-N-GSDMD (#36425S, CST, dilution 1:1000) antibodies at 4 °C over night. After primary antibody incubation blots were washed three times with TBST, HRP-conjugated secondary antibodies (#7074, CST, dilution 1:2000) were probed for 1 h at room temperature prepared in TBST. Finally, blots were washed with TBST for 3 min each at room temperature and developed by ECL (SuperSignal™ West Femto Maximum Sensitivity Substrate, #34096, Thermo Fisher Scientific) in Li-COR imager (Odyssey® XF), Image Studio software. Relative protein levels were quantified using ImageJ software.

**Kidney bulk RNA-seq**. Total RNA was isolated using RNeasy mini kit (Qiagen). Sequencing libraries were constructed using the Illumina TruSeq RNA Preparation Kit. High-throughput sequencing was performed using Illumina HiSeq4000 with 100 bp single-end according to the manufacturer's instruction.

**Kidney bulk RNA-seq data analysis**
*Quality control, alignment, differential expression analysis.* Adaptor and lower-quality bases were trimmed with Trim-galore. Reads were aligned to the Gencode mouse genome (GRCm38) using STAR. Gene and isoform expression levels (TPM) were estimated using RSEM. DEGs between control and disease groups were identified using DESeq2. To examine the enrichment of the DEGs in single cell clusters, a z score of normalized expression value was first obtained for every single cell. Then, we calculated the mean z scores for individual cells in the same cluster, resulting in 30 values for each gene. The z scores were visualized by heatmap showing the enrichment patterns of the genes across the cell types.

*Deconvolution.* Bulk RNA-seq deconvolution was performed using two independent methods, Multi-subject Single Cell deconvolution (MuSiC) and BisqueRNA[72,73]. MuSiC weights genes showing cross-subject and cross-cell consistency, enabling the transfer of cell type-specific gene expression information from one dataset to another. After inputting bulk and scRNA-seq data, bulk cell type proportions were estimated using function *music_prop*. BisqueRNA captures relative abundances of a cell type across individuals. Note that these abundances are not proportions, so they cannot be compared between different cell types. Function *ReferenceBasedDecomposition* with subcommands *markers = NULL*, *use.overlap = F* was used.

**Preparation of single-cell suspension**. Euthanized mice were perfused with chilled 1× PBS via the left ventricle. Kidneys were harvested, minced into approximately 1 mm³ cubes and digested using Multi Tissue dissociation kit (Miltenyi, 130-110-201). The tissue was homogenized using 21G and 26.5G syringes. Up to 0.25 g of the tissue was digested with 50 μL of Enzyme D, 25 μL of Enzyme R, and 6.75 μL of Enzyme A in 1 mL of RPMI and incubated for 30 min at 37 °C. Reaction was deactivated by 10% FBS. The solution was then passed through a 40 μm cell strainer. After centrifugation at 400 × g for 5 min, cell pellet was incubated with 1 mL of RBC lysis buffer on ice for 3 min. Cell number and viability were analyzed using Countess AutoCounter (Invitrogen, C10227). This method generated single-cell suspensions with > 80% viability.

**Single-cell RNA-seq**. 10,000 cells were loaded into the Chromium Controller (10X Genomics, PN-120223) on a Chromium Single Cell B Chip (10X Genomics, PN-120262) and processed to generate single-cell gel beads in the emulsion (GEM) according to the manufacturer's protocol (10X Genomics, CG000183). Libraries were generated using Chromium Single Cell 3' Reagent Kits v3 (10X Genomics,

PN-1000092) and Chromium i7 Multiplex Kit (10X Genomics, PN-120262) according to the manufacturer's manual. Quality control for constructed library was performed by Agilent Bioanalyzer High Sensitivity DNA kit (Agilent Technologies, 5067-4626) for qualitative analysis. Quantification analysis was performed by Illumina Library Quantification Kit (KAPA Biosystems, KK4824). The library was sequenced on Illumina HiSeq 4000 system with 2 × 150 paired-end kits using the following read length: 28 bp Read1 for cell barcode and UMI, 8 bp I7 index for sample index, and 91 bp Read2 for transcript.

**Single-cell RNA-seq data analysis**
*Alignment and quality control.* Raw fastq files were aligned to the mm10 (Ensembl GRCm38.93) reference genome and quantified using CellRanger. Seurat was used for data quality control, preprocessing, and dimensional reduction analysis. After gene-cell data matrix generation of $n = 12$ IRI samples and $n = 6$ control samples, all 18 matrices were merged and poor-quality cells with < 200 or > 3000 expressed genes and mitochondrial gene percentages > 50 (to take into account high mitochondrial content in proximal tubule cells) were excluded. Remaining barcodes of high-quality cells were exported.

*Batch integration.* 10X filtered output matrices of all 18 batches were merged and subset to the barcodes of cells above and subjected to batch integration using LIGER (Linked Inference of Genomic Experimental Relationships) package[36]. In short, LIGER identifies shared and dataset-specific factors through integrative non-negative matrix factorization (iNMF). After normalization by number of unique molecular identifiers (UMIs), gene expression was scaled but not centered because NMF requires non-negative values. Function *optimizeALS* was used with kappa = 20 and lambda = 5, followed by quantile normalization using function *quantile_norm* as well as joint clustering with *louvainCluster* with a resolution of 0.25. For dimensionality reduction, dimensions 1:ncol(object@H.norm), perplexity = 30 and theta = 0.5 were used as per default settings for visualization in UMAP space. Function *ligerToSeurat* was used to export the results back to Seurat.

*Ambient RNA quantification.* As in droplet-based scRNA-seq experiments there is always a certain amount of background mRNAs present that get distributed into droplets and sequenced along with cells. In order to quantify the net effect of ambient RNA contamination, we used R package SoupX[74]. Function *autoEstCount* was used to estimate the contamination fraction in P0 and adult batches separately. We visualized the change in expression due to soup correction in UMAP space. Function *adjustCounts* was used to correct the count expression matrices for downstream processing. We then used the corrected matrices and reran the whole Seurat pipeline with the same parameters. The average expression of genes per cluster were used for Pearson correlation coefficient analysis to compare between matrices with and without ambient RNA correction. As results with and without ambient RNA were similar, results without ambient RNA correction are shown throughout the manuscript.

*Removal of doublet-like cells.* Doublet-like cells were identified using package DoubletFinder[75]. Assuming no ground truth in order to facilitate an unbiased approach, pK was identified using *paramSweep_v3* function with PCs = 1:10. Homotypic doublet proportion was estimated with function *modelHomotypic* using above clustering information after LIGER integration as annotations. Finally, function *doubletFinder_v3* was run with pN = 0.25, pK, and nExp as identified by the functions above. After excluding 9394 cells that were identified to likely be doublets, 113,579 singlet cells remained.

*Identification of marker genes and differentially expressed genes.* The whole pipeline in Seurat and LIGER was run again on the remaining 113,579 high quality singlet cells to produce the final dataset used for all downstream analyses. Differentially expressed genes in cell clusters were identified in Seurat using *FindAllMarkers* function with parameters test.use = MAST, min.pct = 0.05 and logfc.threshold = 0.2 and a list of marker genes[21] was used for manual annotation of cell types to the 18 identified clusters in the final dataset.

*Proximal tubule cell subclustering analysis.* The whole Seurat and LIGER pipeline was again repeated with only those barcodes of cells annotated as PT cells. The same settings were used for the pipeline as stated above, yielding 11 PT subclusters.

*Immune cell subclustering analysis.* Similarly, all cells belonging to clusters annotated Granul_1, Granul_2, Mono_1, Mono_2, Macro, T cell_1, T cell_2, and B cell were subjected to subclustering analysis, yielding 17 immune cell subclusters.

*Cell cycle analysis and IRI scoring.* Cell cycle scoring was performed with Seurat package using function *CellCycleScoring* with cell cycle gene sets provided by Kowalczyk et al.[76]. To keep biological information intact, cell cycle genes were not regressed out from the dataset. Similarly, we customized the same function to create an IRI score using two gene sets for cells from Control and IRI samples, respectively. Gene sets were derived by running differential gene expression analysis between Control and IRI samples using *FindAllMarkers* function in Seurat. The 100 most specific DEGs for both conditions were chosen and subjected to

Pearson correlation analysis, pruning genes with poor auto- or anticorrelation, thereby ensuring high intra-group correlation and high inter-group anticorrelation. The underlying scoring strategy enabling us to classify cells into either Control, IRI or neither was described by Tirosh et al.[77].

*scRNA-seq trajectory analysis*
Slingshot: To construct single cell pseudotime trajectories of PT cells and to identify genes whose expression changed as the cells underwent transition, package Slingshot[78] was applied to a random sample of PT cells consisting of an evenly distributed 150 cells per PT subcluster, resulting in a total of 1650 cells. First, genes were filtered for cell type markers with at least 3 reads in at least 10 cells. Next, counts were normalized and dimensionality was reduced using diffusion maps with package destiny[79]. Then, cells were clustered with Gaussian mixture modeling (GMM) making use of package mclust[80]. Slingshot functions *getLineages* and *getCurves* were used to calculate trajectories. In order to identify temporally differentially expressed genes, generalized additive modeling (GAM) was applied with a locally estimated scatterplot smoothing (LOESS) term for pseudotime. The top genes were picked based on $p$ value and their expression over pseudotime was visualized in a heatmap.

Monocle2 & Monocle3: Slingshot-derived pseudotime trajectories were validated with Monocle2[81] and Monocle3[82] packages using the same cells as input. Genes for ordering cells were selected if they were expressed in ≥ 10 cells, their mean expression value was ≥ 0.05 and dispersion empirical value was ≥ 2. Highly variable genes along pseudotime were identified using *differentialGeneTest* function of Monocle2 with $q < 0.01$. Individual branches were analyzed using *BEAM* and *plot_genes_branched_heatmap* functions. In Monocle3 cells were re-clustered using a resolution of $3e^{-3}$ with k-nearest neighbor (kNN) $k = 29$. The trajectory was produced using default parameters of function *learn_graph*. Cluster centers of samples harvested 1d after ischemia were set as root node before ordering cells along pseudotime with function *order_cells*.

TradeSeq: For further analysis of gene expression along trajectories, package tradeSeq was used downstream of slingshot. First, a negative binomial model was fitted after an optimal number of knots was estimated. Next, general additive models were fitted with function *fitGAM* using nknots = 5, as determined by *evaluateK* function. Gene expression was projected onto coordinates in a two-dimensional space, as provided by Slingshot, using function *plotGeneCount*. Function *plotSmoothers* was used to visualize gene expression along the 2 PT trajectories over pseudotime.

RNA velocity: To calculate RNA velocity, Python-based Velocyto command-line tool as well as Velocyto.R package were used as instructed[83]. We used Velocyto to calculate the single-cell trajectory/directionality using spliced and unspliced reads. From loom files produced by the command-line tool, we subset the exact same cells that were previously selected randomly for Slingshot trajectory analysis. This subset was loaded into R using the SeuratWrappers package. RNA velocity was estimated using gene-relative model with k-nearest neighbor (kNN) cell pooling ($k = 25$). The parameter n was set at 200 when visualizing RNA velocity on the UMAP embedding.

*Differential proportion analysis*. Changes in cell population proportions across groups were analyzed as described previously[84]. Briefly, to analyze whether observed changes in proportions of cell populations are greater than expected by chance, a permutation-based statistical test was used. We used the original source code from the authors, so that both technical variation within the experimental technique (e.g., absolute cell numbers within the experiment, cell-type capture bias) and variation due to in silico analysis (e.g., cluster assignment accuracy) were considered. After creating a proportion table of clusters per phenotype/group, the difference in cluster proportion was compared with a null distribution, which was constructed by randomly permuting random subsamples of cluster labels across a random proportion of total cells. This was done 100,000 times and the observed distribution was then compared with the null distribution, from which the final p values were obtained. As suggested in the original paper, we set the w parameter to 0.1, so that lower values would trend toward a stricter test (fewer significant hits), and higher values toward higher numbers of significant hits.

*Gene regulatory network inference*. To identify TFs and characterize cell states, we employed *cis*-regulatory analysis using the R package SCENIC[39], which infers the gene regulatory network based on co-expression and DNA motif analysis. The network activity is then analyzed in each cell to identify recurrent cellular states. In short, TFs were identified using GENIE3 and compiled into modules (regulons), which were subsequently subjected to *cis*-regulatory motif analysis using RcisTarget with two gene-motif rankings: 10 kb around the TSS and 500 bp upstream. Regulon activity in every cell was then scored using AUCell. Finally, binarized regulon activity was projected onto Slingshot-created UMAP trajectories.

*Weighted gene coexpression network analysis (WGCNA)*. We applied WGCNA to our scRNA-seq dataset using the R package WGCNA, as described previously[85,86]. First, to circumvent the sparsity of single-cell data we constructed metacells with a bootstrapped aggregation process to single-cell transcriptomes and pooled cells within the same cell type and experimental group to retain these metadata for WGCNA. We then created a similarity matrix, in which the similarity between genes reflects the sign of the correlation of their expression profiles. To emphasize strong correlations and reduce the emphasis of weak correlations on an exponential scale, we raised the signed similarity matrix to power β. The resulting adjacency matrix was transformed into a topological overlap matrix. Modules were defined using the following specific module-cutting parameters: module size = 50 genes, deepSplit Score = 4, threshold of correlation = 0.2. Modules with a correlation of > 0.8 were merged. The first principal component of the module, the module eigengene (ME), was used to correlate with experimental group. Hub genes were defined using intra-modular connectivity (kME) parameters of the WGCNA package.

*Ligand–receptor interactions*. To assess cellular crosstalk between different cell types, we used the CellPhoneDB repository to infer cell–cell communication networks from single-cell transcriptome data[87]. We used the Python package CellPhoneDB with database v2.0.0 to predict cell type-specific ligand–receptor interactions as per the authors' instructions. Only receptors and ligands expressed in more than 5% of the cells in the specific cluster were considered. 1000 iterations of permutation test were conducted and $p$ values were corrected with FDR methods. We kept only statistically significant means of interaction partners from the CellPhoneDB output, serving as a proxy for mean expression of both ligand and receptor of a given predicted connection. Connections were visualized in Circos plots after quantification using Circos Table Viewer[88].

*Gene set enrichment analysis (GSEA) of LINCS drug database*. To inform about potential treatment candidates and driver pathways of the observed maladaptive kidney response signature, we performed GSEA on drug PRLs derived from the Library of Integrated Network-based Cellular Signatures (LINCS, http://www.lincsproject.org/) drug database, which catalogs transcriptional responses following treatment with small molecules, as described recently[44]. In short, a weight average fold change difference between treated and untreated conditions representing the drug effect was calculated for each gene, and genes were ranked according to their differential expression. Ranked lists for each drug in L1000 Phase 1 and Phase 2 experiments were merged according to the PRL methodology[45], creating drug PRLs representing consensus transcriptional drug response patterns. We then queried PRLs for the maladaptive gene signature derived from PT subcluster DEG analysis, using GSEA[89] to find drugs with positive normalized enrichment scores (NES), indicating upregulation of the maladaptive signature, and negative NES, indicating downregulation of the maladaptive signature. Bonferroni-corrected $p$ values < 0.05 were used.

**Statistics & reproducibility**. Data are expressed as means ± SEM unless otherwise stated. Statistical analyses are indicated in the respective "Methods" section and figure legends. Appropriate parametric or non-parametric tests were performed as per normality distribution. $P < 0.05$ was considered to be statistically significant. No statistical method was used to predetermine sample size. No data were excluded from the analyses.

**Reporting summary**. Further information on research design is available in the Nature Research Reporting Summary linked to this article.

## Data availability
Raw and metadata are available at GEO accession number "GSE180420". Processed data are available via an "interactive website [www.susztaklab.com/ischemia_reperfusion_injury/scRNA/]". All other relevant data supporting the key findings of this study are available within the article and its Supplementary Information files or from the corresponding author upon reasonable request.

## Code availability
Codes to reproduce all parts of the analysis are provided via a "GitHub repository [https://github.com/ms-balzer/IRI_adaptive_maladaptive_kidney_regeneration]"[90].

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

## Acknowledgements
Work in the Susztak lab is supported by the NIH DK076077, DK087635 and DK105821. M.S.B. is supported by a German Research Foundation (Deutsche Forschungsgemeinschaft, DFG) grant (BA 6205/2-1). We thank the University of Pennsylvania Diabetes Research Center (DRC) for the use of the next generation sequencing Core (P30-DK19525).

## Author contributions
M.S.B. and K.S. designed and conceived the experiment. M.S.B., Z.M., and R.S. conducted the single-cell experiments. M.S.B., T.D., and Y.-W.Y. conducted IRI surgery. D.L.A. conducted flow cytometry with help from M.S.B. and supervision from C.A.H. M.B.P. conducted histopathological analysis. M.S.B. conducted immunofluorescence, cell culture, and qPCR analyses. D.M. conducted Western blotting. M.S.B. conducted bioinformatics analysis with help from H.H. and H.M. All authors discussed and commented on the results. M.S.B. and K.S. wrote the manuscript and all authors edited and approved of the final manuscript.

## Competing interests
The authors declare no competing interests.
