## [Peer Review File · Nature Communications]

Single-cell analysis highlights differences in druggable pathways underlying adaptive or fibrotic kidney regenerationREVIEWER COMMENTS

Reviewer #1 (Remarks to the Author):

This study by the Susztak lab is generally well performed and interesting. The authors provide a huge bioinformatical dataset obtained from a series of simple experiments with a limited number of mice (n=6 in some cases). This referee has not seen an RNA analysis in this much detail on proximal tubules before. In addition, they confirm a previously published role of ferroptosis, an iron dependent cell death. Finally, a compound-based pathway targeting screen is added, all in all rendering this a very comprehensive study. However, several aspects should be revisited. Most importantly, the story as it stands is quite confusing and a central message is unclear.

Major concerns

- Lots of data are presented, and lots of conclusions are indicated in the results section. However, a central point should be more clearly defined rather than simply lining up bioinformatical data.
- It was conclusively published years ago that in kidney IRI, ferroptosis drives proximal tubular damage (Ref 52-56?). Why is it surprising that the FA oxidation/ OxPhos systems are the major drivers of PT injury in IRI – is this just another way of describing the same thing?
- The evidence for GSDMD in this paper is very limited. I would not recommend to conclude anything on pyroptosis here. To be conclusive, cleaved GSDMD should be demonstrated on protein level in isolated proximal kidney tubules (in which macrophages can be excluded to contaminate the system).
- Why do the authors believe that their model compounds in their screen are at all on target? The possibility for any off-target effect should be discussed for each compound used in the “screen”. For VX765, a complete off-target evaluation is required if the authors insist on concluding on what they refer to as pyroptosis. In addition, bc of several amines in the structure of VX765, it should be investigated as an inhibitor of “ferroptosis”.
- The authors selected some terms from the KEGG pathway analysis in Fig. 4. For these terms highlighted in “yellow”, it would be important to demonstrate directly within this publication what actually identified them. In other words, what was measured to make these terms come up and would these primary terms allow to conclude on e.g. apoptosis, ferroptosis, etc.
- In Fig. 4b, data on pyroptosis are presented. IFN and IL-1b expression are no indication for pyroptosis to actually happen. No big changes are demonstrated for the major key player GSDMD. IL-18 is not even mentioned. DDx58 is not important in pyroptosis. I cannot see why this pathway should be mentioned at all. Likewise, SAT1 and Map1lc3B are not known as important players in ferroptosis. Even more striking, apart from the caspases and the BCL2 proteins, not of these proteins are important in apoptosis. Who made these very confusing lists?
- Erlotinib and crizotinib must be checked for off-target effects on ferroptosis.

Minor remarks

- The abstract contains individual paragraphs – I am not sure if this is appropriate for NatComm.
- This referee assumes that the IRI model was performed in a strictly double blinded manner. If so, the blinding strategy should be added to the material and methods section in the IRI description.
- How would changes in RNAs at all allow to conclude on renal function in any sense? Do RNAs have direct effects on signaling pathways? If not, why not show protein expression?
- In Fig. 7D – what defines the “immune cell fraction”?
- Fig. 7E- differences in the PT7 population are entirely correlative with IRI, they by no means indicate this population to be mechanistically involved in the maladaptive repair phenotype.

Reviewer #2 (Remarks to the Author):

The manuscript by Balzer et al. details a single cell transcriptomic study of acute kidney injury (AKI) and chronic kidney disease (CKD).

Single-cell and bulk RNA-seq was performed on 2 mouse kidney samples after short and long IRI at different time points – day 1, day 3 and day 14 after ischemia.

Bulk RNA-seq is informative in showing overall changes in gene expression while the single-cell data – from a total of 113,579 cells – in general is used to show population changes between conditions. There is a marked increase in immune cells following injury, with the emergence of a maladaptive proximal tubule cell type seen in the long IRI exclusively at day 14. The authors describe how this population possibly recruits immune cells to the kidney and demonstrate that inhibitors of a relevant biological pathways can interrupt this process in long IRI.

This is a substantial dataset and it has been comprehensively analysed to produce a biologically relevant observation. However, there are a number of things I think need a significant amount of clarification and discussion in the manuscript.

Major comments

It is somewhat difficult to link the experimental design with the analysis of the single cell experiments – in particular the kinetic aspects. Samples were taken at three time points after injury – day 1, day 3 and day 14, and yet the data is generally presented as control, short (injury) and long (injury), where I gather the three time points have been amalgamated (e.g figure 1j). This is not really made explicit in the text, which makes it a little bit confusing but perhaps more importantly gives the sense that a major aspect of the experimental design is not explored. I see in the supplementary figures there is reference to the cell population sizes over the time series (e.g. figure S1d and e) but figure S1d is really hard to interpret, especially the time post-op figure (these figures are tiny and very detail rich) but understanding the kinetics of the response seems like something that should be far more visible in the manuscript, and merging datasets like this without accounting for experimental design runs the risk of obscuring the biology and mixing effects (time after injury and treatment) – in the very least a more detailed discussion of why this was done is needed, and many of the figures in supplementary would be more helpful if presented in the main figures (e.g. figure 1e)

The definition of the maladaptive/normal PT cells is also needs some clarification. What are the marker genes used to define the PT population in total? And what proportion of the PT cells are derived from the control and treated animals?

It is unclear from the data shown if the maladaptive cells express PT cell markers – how similar are these cells to the other PT cells? Are they the same cell type that express an additional cassette of genes or are they fundamentally different in some way?

From Figure 2h it seems the maladaptive cells cluster as granulocytes when integrated with another scRNA-seq dataset.

If the hypothesis is that the maladaptive cells emerge from normal PT cells, this needs to be stated explicitly – this informs whether or not the pseudotime analysis is a meaningful pursuit – if the cells are from two different lineages then one would not expect an intermediate population. Following on from this, are there any intermediate cells evident in the early time points (1 / 3 days)?

Minor comments

- The term “IRI” is critical to understanding the manuscript but I could not see it defined anywhere

- “In addition, few if any studies performed follow-up validation experiments confirming the role of specific pathways.” This statement is ambiguous and should be revised for clarity. There are either 0 or >0 studies that do this. If there are studies with follow-up

validation they should be cited and discussed here.

- According to the methods ambient RNA quantification is performed but not used throughout the manuscript. The results section states that cell filtering involved "ambient RNA and batch effect correction". This should be clarified.
- In general the single-cell data quality seems somewhat low, with what appears to be >1000 genes per cell in most cells, and overall high mitochondrial counts (and a 50% cutoff). While this is clearly sufficient to differentiate the different cell types, it does seem a little unusual – do the authors anticipate high mitochondrial load in some cell types (we have seen this in other tissues), and in general how does the gene detection in this dataset compare to other datasets from the same tissue.
- I was unable to access the interactive dataviewer but I guess this is embargoed until publication.
- Supp figure 1 – samples labelled both norm and control in adjacent figures, there is a sample labelled scl, it's not clear what this is.
- Supplementary table 3 – the rows are not defined

Reviewer #3 (Remarks to the Author):

In this paper, Balzer et al developed a model of adaptive and fibrotic kidney regeneration by titrating IRI. The authors identified proximal tubule cells as being susceptible to injury using sc-RNA seq. They also identified that repair is correlated with fatty acid oxidation and oxidative phosphorylation. They also identified a signature of fibrotic (maladaptive) response in long ischemia exp. and myeloid chemotactic signature. Using druggability analysis, the authors identified pyroptosis/ferroptosis as a maladaptive response.

This paper is of interest to the scientific community: trying to identify new pathways that can be targeted to prevent fibrotic responses are important to preserve kidney function.

It is valued the effort of defining two different types of IRI and performing bulk and sc-RNA seq analysis in these samples and determining the molecular signatures of the involved in the fibrotic processes. It is also recognized the expected complexity of analyzing these data and the efforts in generating these animal models.

Nevertheless, the work is lacking in some important aspects, that in the end limit the novelty and significance of this work.

Generally speaking, the methods are not clear and are missing a lot of details (hard to determine reproducibility of the data), especially because only two samples per time point are analyzed. With the variability of inducing an IRI injury in an animal model, it should be taken into consideration that two samples are a very limiting factor when important processes need to be established (or, like in this case, determining differences between a short time interval). It is recognized that doing more samples per time point might be hard, but the authors failed to provide information on the validity of their model (more physiological and histological data would help with the model validation, see comments below).

Another important point is that pyroptosis/ferroptosis (as well as myeloid attraction) injury are not a novel concept in the context of IRI. Quite a few papers evidence these pathways. It would be important for the authors, to increase novelty to their work, clearly describing the differences with the already published work.

Major comments:

1. As stated above the number of samples (#2) for the time points considered is a very limiting factor of the analysis, especially in the context of IRI (induced damage might be quite variable)
2. Please clearly explain the rationale of 23' minutes vs 30'? Are 7 minutes enough to determine reversible vs irreparable damage?
3. Fig. 1b. No statistical significance is reported by the BUN between long and short term at 14d. How many mice were used in Fig.1b? It is advised to perform BUN and creatinine at least in 10 mice per group, not only in the animals that were used for bulk and sc-RNA

seq (this will give more support to the 2 samples/time point analysis per bulk and sc-RNA).

4. Fig 1b. Parameters are measured at 3 and 14 days but it is unclear if something happens in between. Why those two time points were chosen without intermediate checkpoints, particularly for functional data that can be taken without sacrificing the animals?

5. From Fig. 1d it seems that also short ischemia develops fibrosis. Was a quantification of fibrosis determined? The differences between long and short might not be so clear. The authors fail to clearly distinguish them. It would be also important to perform staining for immune cells. A lot is mentioned about immune markers, but no cell infiltration differences are shown between the groups (for example by staining and quantification).

6. It is not clear if bulk and sc-RNA were done on the same kidney.

7. Fig. 1g. MMP9 could be overexpressed by many cells following injury: how reliable is this marker to identify granulocytes in this specific experiment?

8. UMAP in Fig. 1f is hard to understand. What is the rationale for combining all the samples together at all the time points? It would be interesting to see the UMAP and analysis for the samples separated by long and short ischemia and vs different time points. The same concept is also true for Fig. 1h-i. It is missing the description of the rationale to explain how the different experimental groups were arranged.

9. In Fig. 2b a maladaptive cluster was identified by combining all the PT cells. It is not clear based on what this cluster was defined as maladaptive.

10. Fig. 2c should follow Fig. 2b; hard to follow the figure (minor comment).

11. In Fig. 2c it is evident in the short IRI a cluster on top that was defined by the authors as injured (this cluster is not present in healthy, nor long IRI). Are these cells injured also maladaptive? It is not clear if a comparison between this injured cluster in the short IRI was compared with the maladaptive cluster in the long IRI.

12. Fig. 3-4. It is unclear how "maladaptive" is defined? Based on what was the maladaptive regeneration distinguished by normal injury and regeneration?

13. In Fig. 3B group legend is missing the colored dots to clearly identify which group belongs to which cluster.

14. The authors are not discussing what could be the changes in the interval between the two time points analyzed.

15. In Fig. 4 the authors report that ferroptosis is altered in both long and short IRI but short IRI stays only on day 1. Then Fig. 5 is the description of the maladaptive cluster with myeloid cells. It is confusing the connection between Fig. 4 and Fig. 5. Two separate processes? Both these two processes are known to occur during IRI. Can the authors specify the novelty of their work compared to the already published reports about these events in IRI?

16. No patient characteristics for Fig. 6 are reported.

17. The authors identify in Fig. 6b crizotinib and erlotinib as drugs that can prevent fibrosis using their data but there are no follow-up experiments. It is not clear the rationale for finding these drugs. Were these drugs tested to determine if their activity? In table Fig. 6C, the authors claim that these drugs induced pyroptosis. Therefore, it is not clear to the reader the description of the role of these two drugs. The logical connection of the investigation of the human samples in Fig. 6 is missing based on the drug analysis done in the first part of Fig. 6.

18. In Fig. 7 the authors use two drugs that inhibit pyroptosis and ferroptosis (they seem to not be connected with the drugs described in Fig. 6b. This point is not clear). Should these drugs be tested in the short IRI since on day 1 also the ferroptosis pathway is present in the short ischemia?

19. Fig. 7k needs more quantification (fibrotic scoring) to convince that the treatment improved structure and is efficient in reducing fibrosis in these animals. In addition, longer time points (past the 14 days, at 2 months, for histology and functional data) should have been provided to confirm that blocking (or preventing) these two pathways (pyroptosis and ferroptosis) rescue kidney function and structure and not just slow down progression.

20. No evaluation/correspondence of genes identified in their IRI models in the human samples. That would have strengthened the paper significantly.

On the following 16 pages we addressed comments by the reviewers. Here we would like to highlight the main changes.

1. We clarified the main message of the work. We developed a model of adaptive and maladaptive/fibrotic kidney regeneration and directly compared single-cell transcriptomic profiling in these models. We define **adaptive** and **maladaptive** (or profibrotic) repair of kidney tubules. We show that severe acute kidney injury induced inflammatory cell death (**ferroptosis and pyroptosis**), and this maladaptive repair attracted immune (myeloid) cells and lead to progressive organ fibrosis.
2. We performed extensive validation to support the role of pyroptosis in maladaptive kidney repair.
 - a. We performed *Gsdmd*-specific *in situ* hybridization (Fig. 6).
 - b. We performed immunofluorescence double-staining against GSDMD and PT marker SLC34A1 (Fig. 6).
 - c. We validated the increase in GSDMD in maladapted samples on the protein level by Western blots (Fig. S11).
 - d. We also performed additional *in silico* analysis leveraging unsupervised weighted gene correlation network analysis (WGCNA), which highlighted *Gsdmd* and *Il1b* as hub genes for maladapted PT cells (Figs. 6, S12, S13).
 - e. Finally, we demonstrate the involvement of pyroptosis in PT cells in kidney damage in independent single cell datasets (Fig. S11).
3. Furthermore, we performed comprehensive profiling of immune cells in our short and long ischemia models by gold standard flow cytometry by completely repeating the mouse experiments.
 - a. We show the pronounced influx of lymphoid and myeloid cells 14d after severe ischemia (Figs. 2, S6), mirroring our single cell transcriptomic results.
 - b. We performed immunostaining studies to show the increased influx of CD11B+ cells into severely damaged kidneys (Fig. S2).
4. We experimentally validated the results of the druggability analysis. We cultured primary mouse renal epithelial cells *in vitro* and used cytotoxicity, viability assays, qPCR, and lipid peroxidation quantification assay (BODIPY staining) to demonstrate that the top 2 drug hits (erlotinib and crizotinib) of our druggability analysis resembling the maladaptive PT transcriptomic signature indeed induced pyroptosis, indicating the importance of a pyroptosis signature in our *in silico* druggability analysis (Fig. S18).
5. Finally, we addressed every single request of the reviewers. We organized edited and added to the original figures. The revised manuscript has 2 additional main figures (9 total), 9 additional supplementary figures (21 total), and 4 additional supplementary tables (11 total).
6. The results have been visualized via our interactive website (www.susztaklab.com/ischemia_reperfusion_injury/scRNA/).
7. All code is made available via a GitHub repository (https://github.com/ms-balzer/IRI_adaptive_maladaptive_kidney_regeneration).

Reviewer #1 (Remarks to the Author):

This study by the Susztak lab is generally well performed and interesting. The authors provide a huge bioinformatical dataset obtained from a series of simple experiments with a limited number of mice (n=6 in some cases). This referee has not seen an RNA analysis in this much detail on proximal tubules before. In addition, they confirm a previously published role of ferroptosis, an iron dependent cell death. Finally, a compound-based pathway targeting screen is added, all in all rendering this a very comprehensive study. However, several aspects should be revisited. Most importantly, the story as it stands is quite confusing and a central message is unclear.

We thank the reviewer for his/her important comments.

We clarified the main message of the work. We developed a model of adaptive and maladaptive/fibrotic kidney regeneration and directly compared single-cell transcriptomic profiling in these models. We define adaptive and maladaptive (or profibrotic) repair of kidney tubules. We show that severe acute kidney injury induced inflammatory cell death (ferroptosis and pyroptosis) and this maladaptive repair attracted immune (myeloid) cells and lead to progressive organ fibrosis.

In the revised version we performed extensive validation studies. We validated the role of pyroptosis in profibrotic PT cells, we validated the druggability analysis, and we validated the immune cell changes in the models (see under).

Major concerns

- Lots of data are presented, and lots of conclusions are indicated in the results section. However, a central point should be more clearly defined rather than simply lining up bioinformatical data.

We thank the reviewer for his/her comments. Our goal was to capture the AKI-to-CKD transition at the single cell level. We show that severe, (but not mild), AKI induces inflammatory cell death (ferroptosis and pyroptosis). Maladaptive repair attracts immune (myeloid) cells and leads to progressive organ fibrosis.

- It was conclusively published years ago that in kidney IRI, ferroptosis drives proximal tubular damage (Ref 52-56?). Why is it surprising that the FA oxidation/ OxPhos systems are the major drivers of PT injury in IRI – is this just another way of describing the same thing?

We show that improved fatty acid oxidation and oxidative phosphorylation are necessary for successful PT repair. It seems from our prior studies that the oxphos defect is reversible and can boost recovery. Many different genes and pathways have been shown to be associated with kidney disease and fibrosis, including apoptosis, ferroptosis, pyroptosis, necroptosis, etc. Here we took an unbiased look with *in silico* and *in vivo* analyses and showed the key role of ferroptosis and pyroptosis (see next comment below for several additional validation studies: WGCNA, *in situ* hybridization, immunofluorescence, Western blot, flow cytometry). The metabolic defect is likely upstream of the inflammatory cell death and fibrosis, as we showed in our prior publications (PMID: 33301705, Dhillon et al.; 31474566, Chung et al.).

- The evidence for GSDMD in this paper is very limited. I would not recommend to conclude anything on pyroptosis here. To be conclusive, cleaved GSDMD should be demonstrated on protein level in isolated proximal kidney tubules (in which macrophages can be excluded to contaminate the system).

Thank you for this important comment. We performed multiple new experiments to validate our transcriptomic findings:

1. We performed *in situ* hybridization using *Gsdmd*-specific probes and show the increased GSDMD expression in proximal tubules after severe injury (**Fig. 6i**).
2. We also performed immunofluorescence double-staining confirming GSDMD expression in severely injured PT (**Fig. 6j**).

3. We performed Western blot analysis (**Fig. S11**) in kidney lysates, with which we show the presence of cleaved gasdermin D protein, not just mRNA.
4. To illustrate this point even more, we now demonstrate a pyroptotic signature in PT cells in several independent scRNA-seq datasets of kidney damage (**Fig. S11**): Although not as striking in extent as in our study, we found markers of inflammasome sensors/adapters, canonical and non-canonical GSDMD activators, *Gsdmd* as the pore-forming executioner itself, and *Il18* and *Il1b* as downstream effectors to be enriched in “failed repair PT” cells of a previous IRI dataset (PMID 32571916, Kirita et al.), in PT cells of a rejecting kidney allograft (PMID 32970632, Dangi et al.), and in PT cells of mice conditionally overexpressing Notch in tubules from our lab (unpublished).
5. Finally, we successfully applied weighted gene correlation network analysis (WGCNA) to our dataset, an unsupervised analysis pipeline. After generating metacells of the PT dataset (**Figs. 6a-b**), we retrieved 8 gene modules (**Figs. 6c, S12a**) demonstrating high specificity for IRI degree and timing post-ischemia (**Figs. 6d, S12b**). Pyroptosis markers *Gsdmd* and *Il1b* proved to be important hub genes (**Fig. 6f**) whose relatively high expression in maladaptive PT cells was highly specific (**Figs. 6g-h, S12**), which was consistent with KEGG pathway analysis (**Fig. S13, Supplementary Table 10**).

We could not find an *in vitro* system that would successfully recapitulate mild and severe ischemic injury *in vivo*, we feel that the combination of evidence is striking and clearly points towards gasdermin D involvement in the PT.

• Why do the authors believe that their model compounds in their screen are at all on target? The possibility for any off-target effect should be discussed for each compound used in the “screen”. For VX765, a complete off-target evaluation is required if the authors insist on concluding on what they refer to as pyroptosis. In addition, bc of several amines in the structure of VX765, it should be investigated as an inhibitor of “ferroptosis”. This is a very important comment. We need to separate 2 issues:

First, the druggability analysis was performed not to find a candidate drug, but to orthogonally validate our transcriptomic analysis findings by comparing our novel maladaptive injury signature with transcriptomic patterns elicited by other drugs in the LINCS database that combines >70k experiments. This analysis found 2 drugs (erlotinib, crizotinib) that elicited transcriptomic signatures in cell culture that overlapped significantly with our maladaptive repair signature. We have worded this more clearly in the manuscript.

We now provide additional *in vitro* validation for the upregulation of *Nlrp3*, *Gsdmd*, *Il18*, and *Il1b* in cultured primary renal tubular epithelial cells after exposure to erlotinib and crizotinib (**Fig. S18**). Also note that neither erlotinib nor crizotinib induced a ferroptosis signature, as shown via qPCR and quantification of lipid peroxidation (BODIPY staining) (**Fig. S18**).

Second, regarding off-target effects of VX765, which is an important comment: As the reviewer points out, small molecular inhibitors of ferroptosis or pyroptosis inhibitors are known to have off-target effects. For example, VX765 inhibits mostly Casp-1 and Casp-4, in addition to being a potent ferroptosis inhibitor. Liproxstatin has been shown not only to interfere with other classical types of cell death (PMID 25402683, Friedmann Angeli et al.) but also voltage-dependent anion-selective channel 1 levels restoring GPX4 levels (PMID 31623831, Feng et al.). Others have found that liproxstatin reduces levels of malondialdehyde and ROS (PMID 31682397, Chen et al.) and prevents both RSL3-induced death of primary human renal PT cells and GPX4 deletion-induced AKI (PMID 25402683, Friedmann Angeli et al.). The role of these and other effects need to be considered.

We agree with the reviewer in acknowledging that we cannot fully exclude off-target effects of neither VX765 nor liproxstatin. In this respect, it is interesting to see that at the single cell gene expression level we could not fully distinguish the targets of VX765 and liproxstatin, as mentioned in the discussion. A previous study has hinted at the interdependencies of the necrotic cell death forms (PMID 32302582, Belavgeni et al.) explaining the similar drug response. We agree with the reviewer that the possibility of off-target effects must be acknowledged. A comprehensive compound evaluation is far beyond the limits and expertise of this paper, we discuss these issues in the discussion section. Finally, we agree that future studies, employing, genetic deletion of ferroptosis and pyroptosis pathway genes in tubule cells will be necessary to delineate the individual contributions of separate necrotic cell death pathways, and we now cite a reference that discusses this issue.

- The authors selected some terms from the KEGG pathway analysis in Fig. 4. For these terms highlighted in “yellow”, it would be important to demonstrate directly within this publication what actually identified them. In other words, what was measured to make these terms come up and would these primary terms allow to conclude on e.g. apoptosis, ferroptosis, etc.

Lists of DEGs by IRI degree and time post-IRI (as provided in **Suppl. Table 7**) were used for the KEGG pathway analysis to generate enrichment scores as depicted in **Fig. 5a** (new numbering). As requested, we now provide these results including genes used for pathway enrichment calculation in **Suppl. Table 8**.

- In Fig. 4b, data on pyroptosis are presented. IFN and IL-1b expression are no indication for pyroptosis to actually happen. No big changes are demonstrated for the major key player GSDMD. IL-18 is not even mentioned. DDx58 is not important in pyroptosis. I cannot see why this pathway should be mentioned at all. Likewise, SAT1 and Map11c3B are not known as important players in ferroptosis. Even more striking, apart from the caspases and the BCL2 proteins, not of these proteins are important in apoptosis. Who made these very confusing lists?

We agree with the reviewer that the marker panels might have been confusing. Originally we used the KEGG unbiased database and we did not want to manually include or delete genes. After reviewing the literature, we now use an updated consensus set of markers implicated in pyroptosis (mostly taken from a recent extensive pyroptosis review, PMID: 33776057, Yu et al.): *Aim2*, *Nlrp3*, *Pycard*, *Nod2*, *Naip2*, *Naip5* (inflammasome sensors and adapters), *Casp1*, *Casp4*, *Panx1*, *P2rx7*, *Casp8* (canonical and non-canonical activation of GSDMD), *Gsdmd* (pore forming executioner of pyroptosis), *Il18*, and *Il1b* (canonical downstream effectors). Ferroptosis markers were taken from WP_FERROPTOSIS gene set (M39768), curated by WikiPathways. Apoptosis markers shown were taken from the Gene Ontology annotation gene set for “apoptotic signaling pathway” (GO:0097190) as well as from HALLMARK_APOPTOSIS (M5902), and KEGG_APOPTOSIS (M8492) gene sets curated by the MSigDB team.

- Erlotinib and crizotinib must be checked for off-target effects on ferroptosis.

We very much appreciate the reviewer’s point.

Now we performed additional *in vitro* studies using primary mouse renal tubular epithelial cells and treated them with erlotinib and crizotinib, which were the top targets of an analysis comparing curated cell culture transcriptomic responses (LINCS database) in the maladaptive PT signature in our dataset. Intriguingly, both drugs are known inducers of pyroptosis. After drug titration to determine adequate treatment doses of crizotinib and erlotinib, we evaluated markers of cell death pathways by qPCR (**Fig. S17**). While treatment with ferroptosis inducer erastin produced strong induction of ferroptosis markers as well as lipid peroxidation, erlotinib and crizotinib did not. Conversely, we found *Gsdmd*, *Il18*, and *Il1b* to be upregulated by both crizotinib and erlotinib, while markers for apoptosis and necroptosis did not change significantly. Surprisingly, there are no studies in the literature

describing the relationship of erlotinib/crizotinib and ferroptosis, highlighting that these novel observations will warrant validation in other organs and tissues.

Minor remarks

- The abstract contains individual paragraphs – I am not sure if this is appropriate for NatComm.

Thank you for pointing this out. We have removed the paragraphs to conform to journal style.

- This referee assumes that the IRI model was performed in a strictly double blinded manner. If so, the blinding strategy should be added to the material and methods section in the IRI description.

Surgeries were performed by two investigators who were blinded to the experimental grouping by using a de-identifying numbering scheme. Mice were grouped and each surgery included short and long, control and treated animals. We have added a short paragraph to the methods section.

- How would changes in RNAs at all allow to conclude on renal function in any sense? Do RNAs have direct effects on signaling pathways? If not, why not show protein expression?

Thank you for stressing this point. While we acknowledge that transcriptomic changes alone do not necessarily reflect protein levels, we and others have repeatedly shown that the transcriptome serves as a very good readout, especially in the context of an unbiased analysis at single cell level (PMIDs 32571916, Kirita et al.; 33531352, Abedini et al.; 33301705, Dhillon et al.; 33859189, Miao et al.). In addition, we have performed several additional protein level validation experiments in the revised manuscript (see above: *in situ* hybridization, immunofluorescence, and Western blots for pyroptosis involvement (**Figs. 6, S11**) and flow cytometry and immunofluorescence for immune cell infiltration (**Figs. 2, S2, S6**), respectively).

- In Fig. 7D – what defines the “immune cell fraction”?

We thank the reviewer for pointing this out, we have added respective legends to the panel of **Fig. 9d** (new numbering).

- Fig. 7E- differences in the PT7 population are entirely correlative with IRI, they by no means indicate this population to be mechanistically involved in the maladaptive repair phenotype.

In **Figs. 9e-g** (new numbering) we highlight cell fraction changes of the maladaptive PT7 population. Despite being subjected to long ischemia, for which we have shown maladaptive features and progression towards fibrosis *in vivo*, pharmacologic inhibition of pyroptosis and ferroptosis ameliorated or prevented the expansion of this maladaptive PT population. Future studies shall dissect the causal role of the maladaptive PT sub-phenotypes using genetic perturbation of, e.g., *Gsdmd*, *Il18* and mouse model studies.

Reviewer #2 (Remarks to the Author):

The manuscript by Balzer et al. details a single cell transcriptomic study of acute kidney injury (AKI) and chronic kidney disease (CKD).

Single-cell and bulk RNA-seq was performed on 2 mouse kidney samples after short and long IRI at different time points – day 1, day 3 and day 14 after ischemia.

Bulk RNA-seq is informative in showing overall changes in gene expression while the single-cell data – from a total of 113,579 cells – in general is used to show population changes between conditions. There is a marked increase in immune cells following injury, with the emergence of a maladaptive proximal tubule cell type seen in the long IRI exclusively at day 14. The authors describe how this population possibly recruits immune cells to the kidney and demonstrate that inhibitors of a relevant biological pathways can interrupt this process in long IRI.

This is a substantial dataset and it has been comprehensively analysed to produce a biologically relevant observation. However, there are a number of things I think need a significant amount of clarification and discussion in the manuscript.

Major comments

It is somewhat difficult to link the experimental design with the analysis of the single cell experiments – in particular the kinetic aspects. Samples were taken at three time points after injury – day 1, day 3 and day 14, and yet the data is generally presented as control, short (injury) and long (injury), where I gather the three time points have been amalgamated (e.g figure 1j). This is not really made explicit in the text, which makes it a little bit confusing but perhaps more importantly gives the sense that a major aspect of the experimental design is not explored. I see in the supplementary figures there is reference to the cell population sizes over the time series (e.g. figure S1d and e) but figure S1d is really hard to interpret, especially the time post-op figure (these figures are tiny and very detail rich) but understanding the kinetics of the response seems like something that should be far more visible in the manuscript, and merging datasets like this without accounting for experimental design runs the risk of obscuring the biology and mixing effects (time after injury and treatment) – in the very least a more detailed discussion of why this was done is needed, and many of the figures in supplementary would be more helpful if presented in the main figures (e.g. figure 1e)

We thank the reviewer for this important comment regarding coherent visualization of the underlying study design. For clarity and easier presentation, we opted for a simplified visualization of only 3 groups (Control, IRI short, IRI long) instead of all 7 (Control, IRI short 1d, IRI short 3d, IRI short 14d, IRI long 1d, IRI long 3d, IRI long 14d) to ease understanding.

We do appreciate the desire for explicit representation of all 7 groups, therefore we implemented changes to the figures (new **Figs. S1b-c**) and added supplemental figures (**Fig. S4a-b, S16b**) to satisfy the needs of those readers with more in-depth interest of all 7 separate experimental groups. We enlarged some of the plots mentioned by the reviewer to enhance the clarity (new **Figs. S1b-d, S2a-b, S5**).

Important to note that the data itself was not merged or manipulated at all. We chose to simplify visualization in order to convey a simpler message focusing on the differences between our 2 model systems (IRI short vs. long) and to focus on changes over time at a later stage in the manuscript (trajectory analysis).

The definition of the maladaptive/normal PT cells is also needs some clarification. What are the marker genes used to define the PT population in total? And what proportion of the PT cells are derived from the control and treated animals?

Thank you. We defined maladaptive, injured, and healthy PT cell subclusters by their marker genes, as shown in **Figs. 1h and 3b**, respectively. We provided the entire DEG lists in **Suppl. Tables 2 and 5**, respectively. As stated in the methods section, we first clustered all kidney cells via unbiased

clustering, revealing the S1 and S3 segments of the proximal tubule (PT), respectively. These clusters were characterized by typical PT marker genes such as *Slc5a2*, *Akr1c21*, *Slc34a1*, *Kap*, *Slc27a2*, *Lrp2*, *Acsm2*, etc. (Fig. 1h and Suppl. Table 2). We then performed PT subclustering analysis as shown in Figs. 3a-b, defining healthy, injured and maladaptive PT subclusters by their DEGs (Suppl. Table 5).

PT cell type proportions contributed by healthy control, IRI short and IRI long groups are presented in Fig. 1i. Cell type proportions contributed by all 7 experimental groups (healthy control, IRI short 1d, IRI short 3d, IRI short 14d, IRI long 1d, IRI long 3d, IRI long 14d) are depicted in Fig. S2b. Proportions of these PT subclusters derived from all 7 experimental groups (healthy control, IRI short 1d, IRI short 3d, IRI short 14d, IRI long 1d, IRI long 3d, IRI long 14d) are shown in Fig. 3d for the ischemia titration model. Important to note that maladaptive PT cells were exclusively contributed by IRI long samples.

It is unclear from the data shown if the maladaptive cells express PT cell markers – how similar are these cells to the other PT cells? Are they the same cell type that express an additional cassette of genes or are they fundamentally different in some way?

We thank the reviewer for this important question. Although we highlight the strong expression of fibroinflammatory marker genes of the maladaptive PT cells (Fig. 3b), they do share (lower) expression of typical PT markers (e.g., *Fth1*, *Kap*, *Gpx3*, *Gatm*, *Acsm2*, *Slc34a1*, *Lrp2*, *Slc27a2*), as indicated in the DEG list (Suppl. Table 2). We generated a dot plot for the reviewer to highlight this fact (see on the right). Interesting to note that some PT sub-cluster cells express lower levels of typical PT markers in a smaller fraction of cells, indicating underlying phenotypic differences.

It stands to reason, however, that they still most likely are PT cells, as in the unbiased analysis, they do co-cluster with healthy PT cells and – most importantly – do present expression of typical PT markers. Lower expression of marker genes in injured/phenotypically different/damaged cells of the same type is a typical phenomenon described extensively in previous kidney scRNA-seq datasets (e.g., PMIDs 29622724 Park et al.; 32571916, Kirita et al.; 33176333, Kuppe et al.).

From Figure 2h it seems the maladaptive cells cluster as granulocytes when integrated with another scRNA-seq dataset.

If the hypothesis is that the maladaptive cells emerge from normal PT cells, this needs to be stated explicitly – this informs whether or not the pseudotime analysis is a meaningful pursuit – if the cells are from two different lineages then one would not expect an intermediate population. Following on from this, are there any intermediate cells evident in the early time points (1 / 3 days)?

Thank you. Integration of our dataset with data from a previous scRNA-seq IRI publication (PMID

32571916, Kirita et al.) confirmed co-clustering of injured PT cells (“NewPT1”) from the previous dataset also with granulocytes (**Fig. S8c**) and highest correlation of our maladaptive PT cluster with the “severely injured PT” cluster from Kirita et al. (**Fig. S8d**), validating the maladaptive phenotype in an independent dataset. Both “NewPT1” and maladaptive PT clusters express lower amounts of typical PT markers (see comment above). Furthermore, several independent trajectory analyses reveal that gradual transitions of transcriptomic profiles over pseudotime perfectly track actual time of our harvesting time points (see **Figs. 4a, b** and **Figs. S9e, g, i**). Along those lines, please note we have added the correct legend panel to **Fig. 4b**, which accidentally was cut off in the plot, conclusively highlighting that pseudotime in the PT lineage tree tracks with actual timing from our experiment (1, 3, 14d after ischemia, respectively).

Minor comments

- The term “IRI” is critical to understanding the manuscript but I could not see it defined anywhere
Thank you for noticing, we have added a definition.

- “In addition, few if any studies performed follow-up validation experiments confirming the role of specific pathways.” This statement is ambiguous and should be revised for clarity. There are either 0 or >0 studies that do this. If there are studies with follow-up validation they should be cited and discussed here.
We agree with the reviewer that the above wording was ambiguous. We have corrected the sentence and cite the respective study that performed follow-up validation experiments.

- According to the methods ambient RNA quantification is performed but not used throughout the manuscript. The results section states that cell filtering involved “ambient RNA and batch effect correction”. This should be clarified.

We state in the methods section that “results with and without ambient RNA were similar” and that therefore “results without ambient RNA correction are shown throughout the manuscript.” We also show ambient RNA levels in the supplement (**Fig. S1c**).

- In general the single-cell data quality seems somewhat low, with what appears to be >1000 genes per cell in most cells, and overall high mitochondrial counts (and a 50% cutoff). While this is clearly sufficient to differentiate the different cell types, it does seem a little unusual – do the authors anticipate high mitochondrial load in some cell types (we have seen this in other tissues), and in general how does the gene detection in this dataset compare to other datasets from the same tissue.

We respectfully disagree with the reviewer in this respect. We show that in comparison with other renal scRNA-seq datasets, we have compiled very high-quality data, indeed: With a sequencing depth of 30,903 mean reads/cell, mean nUMI/cell = 3,689, and mean nGenes/cell = 1,206 (**Fig. S1b**), we have a dataset that is comparatively deeply sequenced. Subsequently, this enabled us to identify substantially more genes/cell than others have done for a similar setup in the kidney (e.g., PMID 32571916, Kirita et al.).

Usual quality control cut-offs for genes/cell in renal single cell datasets can be as low as <150 for the lower and as high as >8000 for upper thresholds (PMID 32571916, Kirita et al.), with the latter bearing the potential to include doublets in the dataset. Our thresholds of <200 and >3000, respectively, are therefore chosen comparatively conservatively and are more in line with what has previously been shown systematically as high-quality cells (PMID 26887813, Ilicic et al.). Important to note that the gene number/cell also depends on the mapping strategy, e.g., one usually retrieves more genes with a non-CellRanger mapping approach. We also show here that the distribution of nUMIs and nFeatures/cell is fairly evenly distributed across most clusters, and that mitochondrial percentage is prominently elevated in tubular epithelial cells, as expected:

It has been reported previously that kidney tubule cells are known to have very high percentages of mitochondrial genes (PMIDs 29622724, Park et al.; 33301705, Dhillon et al.; 33859189, Miao et al.), and regarding the mitochondrial cut-off <50%, we have repeatedly demonstrated high-quality information using this threshold, including the first single cell kidney atlas (PMIDs 29622724, Park et al.; 33301705, Dhillon et al.; 33859189, Miao et al.). One needs to bear in mind that this is a single cell, not single nucleus, dataset. In snRNA-seq datasets, mito% cut-offs can be as low as 1-2% due to the fact that mitochondrial genes most likely reflect cytoplasmic contamination. Lowering the threshold for mito percent cut-off in a single cell (not nucleus) RNAseq kidney dataset would therefore increase the risk of losing biologically informative cells.

- I was unable to access the interactive dataviewer but I guess this is embargoed until publication.

We apologize for this issue. It has not been easy to visualize data. We use AWS and it is sometimes slow. The interactive website http://www.susztaklab.com/ischemia_reperfusion_injury/scRNA/ works on our end from multiple computers, even cell phones, and has been online since the submission date. Please note that above URL is not a secured https:// protocol, several internet browsers might block such non-secure http:// connections by default.

- Supp figure 1 – samples labelled both norm and control in adjacent figures, there is a sample labelled scl, it's not clear what this is.

We thank the reviewer. We have corrected this and made the figures consistent. In keeping with an earlier comment of this reviewer, we chose to adapt **Figs. S1b-c** to reflect the experimental setup with 7 groups.

- Supplementary table 3 – the rows are not defined

We apologize, we have corrected this.

Reviewer #3 (Remarks to the Author):

In this paper, Balzer et al developed a model of adaptive and fibrotic kidney regeneration by titrating IRI. The authors identified proximal tubule cells as being susceptible to injury using sc-RNA seq. They also identified that repair is correlated with fatty acid oxidation and oxidative phosphorylation. They also identified a signature of fibrotic (maladaptive) response in long ischemia exp. and myeloid chemotactic signature. Using druggability analysis, the authors identified pyroptosis/ferroptosis as a maladaptive response.

This paper is of interest to the scientific community: trying to identify new pathways that can be targeted to prevent fibrotic responses are important to preserve kidney function.

It is valued the effort of defining two different types of IRI and performing bulk and sc-RNA seq analysis in these samples and determining the molecular signatures of the involved in the fibrotic processes. It is also recognized the expected complexity of analyzing these data and the efforts in generating these animal models. Nevertheless, the work is lacking in some important aspects, that in the end limit the novelty and significance of this work.

Generally speaking, the methods are not clear and are missing a lot of details (hard to determine reproductivity of the data), especially because only two samples per time point are analyzed. With the variability of inducing an IRI injury in an animal model, it should be taken into consideration that two samples are a very limiting factor when important processes need to be established (or, like in this case, determining differences between a short time interval). It is recognized that doing more samples per time point might be hard, but the authors failed to provide information on the validity of their model (more physiological and histological data would help with the model validation, see comments below).

Another important point is that pyroptosis/ferroptosis (as well as myeloid attraction) injury are not a novel concept in the contest of IRI. Quite a few papers evidence these pathways. It would be important for the authors, to increase novelty to their work, clearly describing the differences with the already published work.

We thank the reviewer for his/her comments. In the revised manuscript version, we have made sure that both *in vivo* and *in silico* methods are clearly outlined in the manuscript such as that all necessary parameters to reproduce our results from both *in vivo* and *in silico* studies are described. In addition, we provide all codes necessary to reproduce the analyses via a GitHub repository (https://github.com/ms-balzer/IRI_adaptive_maladaptive_kidney_regeneration).

Single cell analysis datasets often include relatively small biological replicates as the final analysis and the power of the study is more related to the number of analyzed cells and not just to the biological replicates. Here we made comparisons between more than 110,000 cells. For example, PT cells in control and IRI groups generate one coherent cluster enabling a comparison of thousands of cells. We have also examined bulk expression data and data from prior studies. As outlined below in the point-by-point answers, we now include several additional functional and histological data from animals not used for scRNA-seq. This corroborates the fine titration of our model, making us very confident in the robustness of our IRI model and the presented data.

We performed multiple new experiments to validate our transcriptomic findings:

1. We performed *in situ* hybridization using *Gsdmd*-specific probes and show the increased GSDMD expression in proximal tubules after severe injury (**Fig. 6i**).
2. We also performed immunofluorescence double-staining confirming GSDMD expression in severely injured PT (**Fig. 6j**).
3. We performed Western blot analysis (**Fig. S11**) in kidney lysates, with which we show the presence of cleaved gasdermin D protein, not just mRNA.
4. To illustrate this point even more, we now demonstrate a pyroptotic signature in PT cells in several independent scRNA-seq datasets of kidney damage (**Fig. S11**): Although not as striking in extent as in our study, we found markers of inflammasome sensors/adapters, canonical and non-canonical GSDMD activators, *Gsdmd* as the pore-forming executioner

itself, and *Il18* and *Il1b* as downstream effectors to be enriched in “failed repair PT” cells of a previous IRI dataset (PMID 32571916, Kirita et al.), in PT cells of a rejecting kidney allograft (PMID 32970632, Dangi et al.), and in PT cells of mice conditionally overexpressing Notch in tubules from our lab (unpublished).

5. Finally, we successfully applied weighted gene correlation network analysis (WGCNA) to our dataset, an unsupervised analysis pipeline. After generating metacells of the PT dataset (**Figs. 6a-b**), we retrieved 8 gene modules (**Figs. 6c, S12a**) demonstrating high specificity for IRI degree and timing post-ischemia (**Figs. 6d, S12b**). Pyroptosis markers *Gsdmd* and *Il1b* proved to be important hub genes (**Fig. 6f**) whose relatively high expression in maladaptive PT cells was highly specific (**Figs. 6g-h, S12**), which was consistent with KEGG pathway analysis (**Fig. S13, Supplementary Table 10**).

Major comments:

1. As stated above the number of samples (#2) for the time points considered is a very limiting factor of the analysis, especially in the context of IRI (induced damage might be quite variable)

We thank the reviewer for this important comment regarding model validity. Important to note that this manuscript focuses on single cell transcriptomic analyses in >110k cells and the underlying differences in the outcomes of the different ischemia models. As the damage and recovery is not completely synchronous, even one kidney with several thousand cells provides sufficient insight, as single cell analysis datasets gain from comparisons between cells, not between a high number of biological replicates. Here we made comparisons between more than 110k cells, enabling us to compare thousands of transcriptomic states between control and IRI groups.

To meet those concerns for histopathological and biochemical analyses, we have included additional kidney function data from animals that were not used in the scRNA and bulk RNA-seq analysis (**Fig. S1a**). These results show BUN and creatinine levels very consistent with those presented in animals with bulk and scRNA-seq information (**Fig. 1b**). Statistical analysis confirms significant differences in kidney function levels, demonstrating a clear dose-response relationship (**Fig. S1a**).

However, we do acknowledge the reviewer’s point that intrinsic variability and inter-lab reproducibility problems are inherent to the IRI model. The reason for this is that environmental and procedural variables substantially play into the effect sizes of a given model (e.g., environmental temperature, flow vs. non-flow environment, isoflurane vs. non-isoflurane anesthesia, heating pad stability, exact timing of the surgery steps, etc.). However, we established a meticulous protocol in our lab ensuring high intra-lab reproducibility of the model. The animals were always purchased from the same vendor and used at the same age. Taking into account a personnel surgery training period of >1 year in which we also fine-tuned all possible environmental variables, we do feel very comfortable about the reproducibility of our model system. In fact, in order to validate immune cell infiltration using flow cytometry (**Fig. 2**) for the revised manuscript, we repeated our titrated model and achieved very similar injury degrees.

2. Please clearly explain the rationale of 23’ minutes vs 30’? Are 7 minutes enough to determine reversible vs irreparable damage?

Along those lines, it is well-accepted in the renal IRI field that differences only several seconds of ischemia time can have pronounced effect on the model outcome (PMID 21252527, Heyman et al.; 32534037, Shiva et al.; 22993069, Wei et al.). In a 1-year lead-up phase to the experiment, we were able to titrate 23 and 30min as the perfect durations in our hands to achieve regeneration vs. chronic changes.

3. Fig. 1b. No statistical significance is reported by the BUN between long and short term at 14d. How many

mice were used in Fig. 1b? It is advised to perform BUN and creatinine at least in 10 mice per group, not only in the animals that were used for bulk and sc-RNA seq (this will give more support to the 2 samples/time point analysis per bulk and sc-RNA).

Thank you, we do appreciate the reviewer's rigor to inquire about the robustness of our titrated ischemia model.

In the revised manuscript we provide an additional figure in which we have included all animals in which we performed bulk and scRNA-seq analysis as well as additional training animals from the lead-up phase to the study, which were used to corroborate the titration dose (**Fig. S1a**). We demonstrate similar dynamics of renal function in more animals and can confirm statistical significance for differences in BUN level. We feel that the main figure should only include those animals in which RNA-seq studies were performed in order to keep the results consistent and only show data for the analyzed animals. Once again, we want to reiterate that the main focus of our paper is showing difference in kidney outcome (fibrosis vs. repair), not so much the difference of BUN levels (which can be insensitive), and correlating these changes with underlying single cell expression data.

4. Fig 1b. Parameters are measured at 3 and 14 days but it is unclear if something happens in between. Why those two time points were chosen without intermediate checkpoints, particularly for functional data that can be taken without sacrificing the animals?

The goal of our work was not to characterize the different injury models in detail but to compare the single cell transcriptomic responses in models that have differences in outcomes (fibrosis vs. repair). Still, we thank the reviewer mentioning this very interesting point around the cell trajectory analysis. Trajectory analysis helps us understand the past history of cells captured at a later time point. As we nicely demonstrate with a very robust methodology of employing 3 different methods of computational trajectory analysis (monocle2, monocle3, RNA velocity), we were able to bridge this knowledge gap. We nicely demonstrate that our experimental harvesting time points conclusively track with computationally inferred pseudotime (**Figs. 4a-b, Figs. S9e-h**).

5. From Fig. 1d it seems that also short ischemia develops fibrosis. Was a quantification of fibrosis determined? The differences between long and short might not be so clear. The authors fail to clearly distinguish them. It would be also important to perform staining for immune cells. A lot is mentioned about immune markers, but no cell infiltration differences are shown between the groups (for example by staining and quantification). We thank the reviewer for this suggestion. This is an important comment. We performed quantitation of fibrosis in Sirius red-stained tissue sections using an automated and unbiased method in 10 representative images per biological sample. The photos were similar orientation in the same part of the kidney at the same magnification. We used the MRI fibrosis tool and color deconvolution plugin for ImageJ, as now described in the methods section. Quantitation is now shown in **Fig. 1e** and demonstrates a statistically significant increase in fibrosis in animals 14d after long ischemia. We appreciate a slight increase in the short ischemia model at 14d, however this was not statistically significant.

Most importantly we repeated the entire short and long IRI experiments and performed flow cytometry analysis to quantify immune cells. Flow cytometry is considered the gold standard for immune cell subtype analysis. Flow cytometry analysis demonstrated the increase in lymphoid and myeloid cells in our IRI models (**Figs. 2, S6**). For example, CD4 and CD8 T cells as well as macrophages, neutrophils, basophils, eosinophils, mast cells, and dendritic cells showed pronounced infiltration in the long IRI model, especially 14d after injury, tracking with our transcriptomic data.

Furthermore, we stained kidney tissue samples for CD11B, a macrophage marker, and noticed an increase in myeloid cells matching our scRNA-seq and FACS datasets (**Fig. S2c**).

6. It is not clear if bulk and sc-RNA were done on the same kidney.

We can confirm that bulk and scRNA-seq were done in the same kidneys and have added this information in the manuscript text.

7. Fig. 1g. MMP9 could be overexpressed by many cells following injury: how reliable is this marker to identify granulocytes in this specific experiment?

Mmp9 was identified by unbiased DEG analysis as among the most differentially expressed markers for granulocytes. In addition, *Mmp9* has been demonstrated in an independent renal IRI scRNA-seq dataset to be upregulated in macrophages after IRI (PMID 32571916, Kirita et al.). It was also shown to be a marker for neutrophils in the kidney fibrosis model of unilateral ureteric obstruction (UUO) (PMID 31132220, Wang et al.). Also, we see many granulocyte markers among the top DEGs of the 2 respective granulocyte clusters (see **Suppl. Table 2**), such as, e.g., *Retnlg*, *S100a8*, *S100a9*, *Ptprc*, *Itgam*, *Itgax*, *Fcer1g*, *Csf3r*, etc. Taken together, on a whole kidney level, our data indicates that changes in granulocyte proportion is a major contributor to the increase in *Mmp9* expression, for which we see significant increases 14d after long IRI at bulk RNA-seq level (see barplot to the right).

8. UMPA in Fig. 1f is hard to understand. What is the rationale for combining all the samples together at all the times points? It would be interesting to see the UMAP and analysis for the samples separated by long and short ischemia and vs different time points. The same concept is also true for Fig. 1h-i. It is missing the description of the rationale to explain how the different experimental groups were arranged.

The goal of our project was to analyze the relationships between healthy, short IRI, and regeneration, as well as long IRI and fibrotic kidneys that we were able to resolve on a single cell level. Integration of datasets is essential to compare conditions. Such integration ensures that, for example, cells that have undergone successful repair after a mild injury are grouped together with healthy control cells. This information would be lost if one were to analyze groups separately.

We understand the reviewers' desire to better gauge an experimental groups' individual contribution to the dataset. This is visualized best by analyzing cell densities such as in **Fig. 1k** (numbering new) and **Fig. 3c**. Therefore, to satisfy the specific interest readers might have into how cells distribute between the respective experimental groups stratified by both injury degree and time point, we have added **Fig. S4** for this purpose, where we show both simple UMAP dimension reduction plots as well as cell density plots. Especially from cell density plots, it is easy to appreciate that differences between short and long ischemia groups are small at early time points and pronounced at the 14d time point (**Fig. S4b**). To keep the manuscript as streamlined as possible, we chose to visualize samples by only injury degree (short vs. long) in the main figures, because the portrayed information does not change drastically when stratifying by both injury degree and time points. These plots are provided in the supplements.

9. In Fig. 2b a maladaptive cluster was identified by combining all the PT cells. It is not clear based on what this cluster was defined as maladaptive.

Thank you. We call the PT7 subcluster "maladaptive" because it is not present in the adaptive repair dataset. This cluster has a proinflammatory and profibrotic signature, in the differential gene expression analysis (**Suppl. Table 5**). For example, the DEG list for this cluster includes *Cxcl2*, *Il1b*, *Ccl3*, *Tyrobp*, *S100a8*, *S100a9*, *Gsdmd*, and *Tgfb1* as markers of increased expression when compared to all other PT cells. We have made this clearer in the manuscript also.

10. Fig. 2c should follow Fig. 2b; hard to follow the figure (minor comment).

Thank you for noticing, we have labeled the figures in the order as they appear in the manuscript text.

11. In Fig. 2c it is evident in the short IRI a cluster on top that was defined by the authors as injured (this cluster is not present in healthy, nor long IRI). Are these cells injured also maladaptive? It is not clear if a comparison between this injured cluster in the short IRI was compared with the maladaptive cluster in the long IRI.

This is a good question! The injured PT_10 subcluster the reviewer is alluding to was called “injured” based on the top DEGs present in this cluster (**Suppl. Table 5**). The signature included markers such as *Lcn2* and *Havcr1*, representing a more “classical” injury signature in prior publications. These cells are present in both the long and short IRI samples and have been characterized as “injured PT” before (PMID 32571916, Kirita et al.).

As described in the methods, cluster-specific DEGs were calculated between a respective PT cluster and all other PT clusters. Interestingly, the “injured PT” clusters are projected at the very beginning of the PT trajectories (**Fig. S9c**), indicating an early injury signature compared to the “maladaptive” signature we observe at later pseudotime/time. In short, the “injured” and the “maladaptive” signatures we observed in the unbiased analysis are indeed very different from each other. Injured PT clusters overlap with earlier time points, the maladaptive PT cluster overlaps almost exclusively with the late time point after severe injury. We have highlighted this information in the manuscript.

13. In Fig.3B group legend is missing the colored dots to clearly identify which group belongs to which cluster. We thank the reviewer for meticulous reading, we have corrected this (now **Fig. 4b**).

14. The authors are not discussing what could be the changes in the interval between the two times points analyzed.

As outlined above, we were able to bridge the samples using trajectory analyses; this is due to the fact that the cell states (injured, repairing, etc.) are not completely synchronous in the samples. Along those lines, we are convinced that having more time points would not change the conclusion of our study. The most important point is that we compared injury models with 2 different outcomes (fibrosis vs. repair) at the peak of the injury and at the time of adaptive and maladaptive repair.

15. In Fig. 4 the authors report that ferroptosis is altered in both long and short IRI but short IRI stays only on day 1. Then Fig. 5 is the description of the maladaptive cluster with myeloid cells. It is confusing the connection between Fig. 4 and Fig. 5. Two separate processes? Both these two processes are known to occur during IRI. Can the authors specify the novelty of their work compared to the already published reports about these events in IRI?

We thank the reviewer for this important comment. The goal of our study was to characterize adaptive repair and regeneration and maladaptive fibrotic repair. We identify differences in ferroptosis and pyroptosis in maladaptive repair, but not in apoptosis and necroptosis. We show the causal role of ferroptosis and pyroptosis using inhibitors. Furthermore, we provide unbiased single cell datasets for mouse IRI, including immune cells (including FACS-based validation) that were not characterized before.

As expected from such a huge dataset, there are several findings at play happening after mild and severe acute kidney injury, respectively. The novelty of our paper is 1) the unprecedented quality of analysis of the underlying processes of both successful repair (in a short) vs. maladaptive repair (long ischemia injury scenario) in the kidney. 2) We show that not only ferroptosis, but also and to a larger extent pyroptosis play a role in epithelial cell damage, myeloid cell attraction and fibroinflammation leading to sustained damage and thus AKI-to-CKD transition. 3) Finally, in follow-up validation experiments, we used 2 pharmacological inhibitors of ferroptosis and pyroptosis and were able to

demonstrate that inhibition of both ferroptosis and pyroptosis essentially blocks AKI-to-CKD transition despite severe injury.

16. No patient characteristics for Fig. 6 are reported.

Patient characteristics for **Fig. 8** (previously **Fig. 6**) can be obtained from **Suppl. Table 1** of the publication we reference (Ref 44), from which these data were taken. For the reviewer's convenience, we include patient characteristics here (mean \pm SD):

	tubule eQTL
Age	59.87 \pm 12.53
GFR (ml/min/1.72m ²)	74.96 \pm 21.23
Tubulointerstitial fibrosis %	3.47 \pm 1.99
Glomerular sclerosis %	-
RIN	8.2 \pm 0.91
5'-3' BIAS	0.43 \pm 0.12
Mito %	22.83 \pm 5.39
Ribo %	2.42 \pm 0.57
Unique mapped %	0.87 \pm 0.02
Unmapped %	0.02 \pm 0.01
Gender %	56 M, 44 F

17. The authors identify in Fig. 6b crizotinib and erlotinib as drugs that can prevent fibrosis using their data but there are no follow-up experiments. It is not clear the rationale for finding these drugs. Were these drugs tested to determine if their activity? In table Fig. 6C, the authors claim that these drugs induced pyroptosis. Therefore, it is not clear to the reader the description of the role of these two drugs. The logical connection of the investigation of the human samples in Fig. 6 is missing based on the drug analysis done in the first part of Fig. 6.

Thank you for this comment. We performed several analyses to validate this conclusion.

Fig. 8 (previously **Fig. 6**) shows that necrotic cell death is involved in our maladaptive kidney injury model. We validate a) with an orthogonal analysis approach (drug screen) and b) in human kidneys.

We do not claim that crizotinib or erlotinib prevent fibrosis using our data. Based on the LINCS database analysis we found that crizotinib and erlotinib produced transcriptomic changes that resembled those of the maladaptive PT signature. We explain in more detail now in the text that the reasoning behind the druggability analysis screen was 1) to find drugs that elicit a transcriptomic signature similar to that of our novel maladaptive PT and 2) to infer insightful information from the top drugs produced under 1). We highlight crizotinib and erlotinib because they elicit a transcriptomic pattern where top-loading upregulated genes nicely overlap with the maladaptive injury signature. The fact that both drugs are known inducers of pyroptosis was an appealing and tempting insight, prompting us to perform *in vivo* studies using inhibitors of pyroptosis to prevent fibrosis.

Along those lines, for the revised manuscript we now performed additional *in vitro* studies in primary renal tubular epithelial cells, showing induction of pyroptosis, but not ferroptosis, apoptosis, or necroptosis genes by crizotinib and erlotinib (**Fig. S18**), further validating our *in silico* findings. We also show that neither crizotinib nor erlotinib induce ferroptosis (**Fig. S18e**), again highlighting the importance of pyroptosis as the most striking transcriptomic signature that was picked up by our druggability analysis.

18. In Fig. 7 the authors use two drugs that inhibit pyroptosis and ferroptosis (they seem to not be connected with the drugs described in Fig.6b. This point is not clear). Should these drugs be tested in the short IRI since on day 1 also the ferroptosis pathway is present in the short ischemia?

The goal of our study was to identify the mechanism and drug targets for maladaptive (fibrotic) repair after IRI. Our study indicated the role of ferroptosis and pyroptosis. We demonstrated their role using pathway inhibitors. We show that PT cells undergoing pyroptosis and ferroptosis release cytokines and attract immune cells likely responsible for fibrosis development.

19. Fig. 7k needs more quantification (fibrotic scoring) to convince that the treatment improved structure and is efficient in reducing fibrosis in these animals. In addition, longer time points (past the 14 days, at 2 months, for histology and functional data) should have been provided to confirm that blocking (or preventing) these two pathways (pyroptosis and ferroptosis) rescue kidney function and structure and not just slow down progression. Thank you, this is an important comment. We have added fibrosis scores in which we quantified the drug effects on kidney structure (**Fig. 9I**). While we agree that studying fibrosis at later time points might be important to understand the trajectory of our 2 models (successful repair vs. maladaptive fibrosis), we wanted to focus our analysis on the early time points and pathways that might be causative to the AKI-to-CKD transition. We want to reinforce that even as “early” as 14d post ischemia we were able to see considerable fibrotic changes. It would be of interest to study chronic progression over a longer time period. Our aim, however, was to study the AKI-to-CKD transition and not CKD progression. The latter is an interesting project in and of itself and beyond the scope of this manuscript, as it warrants the choice of a different model system better suited for CKD progression analysis.

20. No evaluation/correspondence of genes identified in their IRI models in the human samples. That would have strengthened the paper significantly.

We thank the reviewer for this excellent suggestion. We have added corresponding analyses in our human dataset for the top targets identified in the murine model to be associated with both the successful repair (*Hnf4a*, *Slc34a1*, *Tmem27*, *Cyp4b1*, *Acsf2*) and maladaptive injury signatures (*Il1b*, *Mmp9*, *Cxcl2*, *S100a9*, *Tyrobp*). In **Fig. S19** we consistently show association of these targets with kidney function and fibrosis, thus strengthening the point that our findings translate to human kidney disease development.

REVIEWERS' COMMENTS

Reviewer #1 (Remarks to the Author):

The authors now submitted an extensively revised manuscript of an already high quality initial submission. All concerns raised by this referee have been satisfactorily responded to, I have no further comments.

That said, the authors should be congratulated to a highly significant contribution to our understanding of AKI.

Reviewer #3 (Remarks to the Author):

The authors have adequately addressed my concerns about the manuscript.

With regard to the data quality, I would maintain that a cutoff of 200 genes per cell is still rather low - but accept that this is now fairly standard within the field. The provided reference (PMID 26887813, Ilicic et al.) refers to data generated using the fluidigm C1 platform which has rather different sensitivity to 10X, and indeed is not cited in the main paper. Regardless this relatively low threshold (biologically speaking) speaks more to the technical limitations of the platforms applied rather than the actual quality of the data.

The observation that some cell types have significantly more mitochondrial content is certainly of interest and it would be useful to mention this in the manuscript - this is an overlooked point in many analyses and useful to highlight as a technical note.

Reviewer #4 (Remarks to the Author):

The Authors have satisfactorily answered the comments. No further comments are requested.

Reviewer #1 (Remarks to the Author):

The authors now submitted an extensively revised manuscript of an already high quality initial submission. All concerns raised by this referee have been satisfactorily responded to, I have no further comments.

That said, the authors should be congratulated to a highly significant contribution to our understanding of AKI.

We thank the reviewer for his/her comments.

Reviewer #3 (Remarks to the Author):

The authors have adequately addressed my concerns about the manuscript.

With regard to the data quality, I would maintain that a cutoff of 200 genes per cell is still rather low - but accept that this is now fairly standard within the field. The provided reference (PMID 26887813, Ilicic et al.) refers to data generated using the fluidigm C1 platform which has rather different sensitivity to 10X, and indeed is not cited in the main paper. Regardless this relatively low threshold (biologically speaking) speaks more to the technical limitations of the platforms applied rather than the actual quality of the data.

We agree. Single-cell RNA-seq technology does have limitations, drop-out and low-abundance gene expression being among them.

The observation that some cell types have significantly more mitochondrial content is certainly of interest and it would be useful to mention this in the manuscript - this is an overlooked point in many analyses and useful to highlight as a technical note.

We agree that mitochondrial content is highly cell type-specific and have highlighted a note on high mitochondrial content in proximal tubule cells within the methods section of the final manuscript.

Reviewer #4 (Remarks to the Author):

The Authors have satisfactorily answered the comments. No further comments are requested.

We thank the reviewer for his/her comments.